# The structure of the teleost Immunoglobulin M core provides insights on polymeric antibody evolution, assembly, and function

Mengfan Lyu [1], Andrey G. Malyutin [2,3,7] & Beth M. Stadtmueller [1,4,5,6] ✉

Polymeric (p) immunoglobulins (Igs) serve broad functions during vertebrate immune responses. Typically, pIgs contain between two and six Ig monomers, each with two antigen binding fragments and one fragment crystallization (Fc). In addition, many pIgs assemble with a joining-chain (JC); however, the number of monomers and potential to include JC vary with species and heavy chain class. Here, we report the cryo-electron microscopy structure of IgM from a teleost (t) species, which does not encode JC. The structure reveals four tIgM Fcs linked through eight C-terminal tailpieces (Tps), which adopt a single β-sandwich-like domain (Tp assembly) located between two Fcs. Specifically, two of eight heavy chains fold uniquely, resulting in a structure distinct from mammalian IgM, which typically contains five IgM monomers, one JC and a centrally-located Tp assembly. Together with mutational analysis, structural data indicate that pIgs have evolved a range of assembly mechanisms and structures, each likely to support unique antibody effector functions.

Polymeric (p) immunoglobulins (Igs) play critical roles in immune system function and are presumed to exist in all jawed vertebrates[1]. Despite the prevalence of pIgs in vertebrate species, considerable differences in structure and function are thought to exist. Typically, pIgs are comprised of two to six Ig monomers, each of which contains two heavy chains and two light chains that form one fragment crystallization (Fc) and two antigen-binding fragments (Fabs). The heavy chain includes three to four constant domains (CH1-CH4) and the Fc region is typically formed by the two C-terminal domains (e.g. CH3 and CH4). The Fc is critical for pIg assembly and for binding to Fc receptors (FcRs) to elicit downstream functions[2,3]. Only a subset of Ig heavy chains are assembled into pIgs, including IgA and IgM in mammals and most birds and reptiles, IgX and IgM in amphibians, IgT and IgM in teleost (t), or bony fish, and IgM in cartilaginous fish. To assemble pIgs, plasma cells typically link multiple copies of heavy chains together with a protein called the joining chain (JC); however, incorporation of the JC is variable among species and heavy chain class[4–6].

In mammals, polymeric forms of IgA are typically dimeric (d), containing two IgA monomers and one JC; however functional trimeric, tetrameric and pentameric pIgA have been identified in lower abundance[7–9]. Polymeric forms of IgM are typically pentameric, containing five IgM monomers and one JC, however, hexamers lacking a JC have been identified in serum[10–12]. JC is necessary for pIgA and pentameric IgM assembly and required for their secretion into the mucosa by the polymeric Ig receptor (pIgR). The pIgR is a transcytotic Fc receptor expressed on the basolateral surface of epithelial cells, which binds to JC-containing pIgA and pIgM and transports them to the apical surface of the cell[13]. On the apical surface the pIgR ectodomain, called secretory component (SC), is proteolytically cleaved, releasing a complex containing SC and the pIg, which is termed a secretory (S) Ig. Whereas pIgM functions in serum and SIgM functions in the mucosa, most pIgA is delivered to the mucosa and functions as SIgA[14]. The pIgM, SIgM and SIgA exhibit unique capabilities compared to monomeric Ig, including high avidity antigen binding, antigen coating or

[1]Department of Biochemistry, University of Illinois Urbana-Champaign, Urbana, IL 61801, USA. [2]Division of Biology and Biological Engineering, California Institute of Technology, Pasadena, CA 91125, USA. [3]Beckman Institute, California Institute of Technology, Pasadena, CA 91125, USA. [4]Department of Bio-medical and Translational Sciences, Carle Illinois College of Medicine, University of Illinois Urbana-Champaign, Urbana, IL 61801, USA. [5]Carl R. Woese Institute for Genomic Biology, University of Illinois Urbana-Champaign, Urbana, IL 61801, USA. [6]Center for Biophysics and Quantitative Biology, University of Illinois Urbana-Champaign, Urbana, IL 61801, USA. [7]Present address: Takeda Pharmaceuticals, Cambridge, MA 02139, USA. ✉e-mail: bethms@illinois.edu

crosslinking, and binding to unique subsets of host and microbial FcRs. The functional outcomes of these interactions are diverse, ranging from complement activation (by IgM), pathogen agglutination and elimination, to commensal microbe colonization and poorly characterized FcR-dependent processes[15,16].

While mammalian pIgs have been studied extensively, less is known about pIgs from other jawed vertebrates. Of particular interest are pIgs from teleosts, which represent an early evolutionary stage of vertebrate adaptive immunity. Moreover, whereas the vast majority of jawed vertebrates, including in cartilaginous fish and amphibians, encode JC in their genomes, teleosts have lost the JC gene during evolution and express polymeric forms of tIgT and tIgM that are presumed to contain four copies of each respective monomer[6,17,18]. Whereas both tIgT and tIgM have been found in fish serum and mucus, tIgT is the predominant mucosal antibody, functionally similar to IgA in mammals[18]. Both tIgT and tIgM can be transported to the mucosa by tpIgR, which has a distinct domain organization and structure compared to mammalian counterparts and does not require a JC to bind tIgT or tIgM[19]. The tSC has been isolated in complex with mucosal tIgT and tIgM, indicating that similar to mammalian SC, tSC remains bound to secretory forms of tIgT and tIgM and may play a role in mucosal immune functions[18,20]. Similar to mammalian pIgs, tpIgs have been associated with complement activation and bacterial coating[21].

The structural mechanisms underlying tpIgs' functions, and more broadly, the evolution of pIg structure-function relationships across species, are poorly understood in-part because until now, teleost antibody structures remained unreported. Furthermore, only in the past several years have structures of mammalian pIgs been reported. Cryo-electron microscopy (cryo-EM) structures of mammalian dIgA, SIgA, and SIgM revealed a central role for the JC, which folded together with C-terminal peptide sequences termed tailpiece (Tps) from the IgA or IgM heavy chains[22–24]. In these structures, both the Tps and the JC contribute β-strands to a β-sandwich near the center of the molecule, which stabilizes the assembly. Furthermore, the JC forms additional contacts with two Fcs, provides part of the pIgR (and SC) binding site, and appears to contribute asymmetry to pIg complexes that otherwise contain multiple heavy chains with identical sequence. Given the critical role for the JC in mammalian pIg structures, we hypothesized that tpIgs lacking JC would assemble using different mechanisms and adopt distinct structures. To test this hypothesis and to explore the evolution and diversity of vertebrate pIg structure and function, we determine the cryo-EM structure of the tIgM Fc core at 2.78 Å resolution. Together with comparative and mutational analysis, this structure provides a glimpse of both a teleost pIg and a pIg lacking a JC, revealing distinct modes of assembly compared to mammalian pIgs and suggesting alternative mechanisms for interacting with antigens and FcRs, including tSC.

## Results

### Overall structure of tetrameric teleost IgM core

Guided by previous structural studies indicating that the human IgM constant domain CH2 and Fabs are flexible and poorly resolved in cryo-EM maps[24,25], we expressed and purified a truncated teleost IgM heavy chain consisting of CHμ3, CHμ4, and the Tp (Fig. 1a, Supplementary Fig. 1). We utilized this tIgM Fc (tFcμ) to determine a cryo-EM structure to an average resolution of 2.78 Å (Fig. 1, Supplementary Fig. 2). In the structure, most main chain and side chain atoms in CHμ4 domains and the Tps were well resolved (resolution range 2.4–3.0 Å). Map resolution was ~3.2 Å and above for CHμ3 allowing the positions of main chain atoms and ~50–75% of side chain atoms in each individual chain to be refined; other side chains were built in geometrically reasonable positions (Supplementary Fig. 2e).

The refined tFcμ structure contains four tFcμ monomers (Fcμ1 to Fcμ4) that lay approximately flat on one plane and include interfaces between Fcμ1 and Fcμ2, Fcμ2 and Fcμ3, Fcμ3 and Fcμ4. Fcμ1 and Fcμ4 are

separated by Tp residues, which form a β-sandwich (hereafter termed *Tp assembly*) comprising two β-strands from each of the four Fcμs (Fig. 1). Overall, the tetramer is ~164 Å across in the largest dimension and can be described as two structurally and symmetry-related halves, one containing Fcμ1-Fcμ2 and associated Tps, and the other containing Fcμ3-Fcμ4 and associated Tps; at the center of the tetramer is a solvent accessible hole with a diameter of ~12.2 Å. Despite similarity between the two halves of the tetramer, the complex is not symmetrical with respect to all four Fcs; rather, adjacent tFcs are related by distinct angles measuring 81°, 72°, 86°, for Fcμ1 and Fcμ2, Fcμ2 and Fcμ3, Fcμ3 and Fcμ4, respectively. Fcμ1 and Fcμ4 are related by an angle of 121° and contact the Tp assembly (Fig. 1b, c).

### tFcμ is formed from Fc monomers containing CHμ4 domains that adopt one of two structural folds

Given the lack of published structural data for a teleost antibody, we first analyzed the structures of tFcμ monomers and their individual domains, CHμ3 and CHμ4. The tFcμ monomers comprise two copies of CHμ3 and CHμ4, which together form a canonical Fc characterized by an angle of ~87° between CHμ3 and CHμ4 on the same chain and interfaces between adjacent CHμ3 and adjacent CHμ4 domains on different chains (Fig. 2a, Supplementary Fig. 3a). Each domain adopts an Ig-constant fold consisting of two β-sheets linked by a disulfide bond, one sheet containing four β-strands termed A, B, E, and D, and the other sheet containing three β-strands termed C, F, and G; the Tp β-strand follows the CHμ4 G strand. In one chain of each Fcμ monomer (A, D, E, and H) the A strand extends several residues, which we have termed A'. Ordered carbohydrates were apparent on CHμ4 residue Asn 374. Within each tFcμ, two CHμ4 domains interact primarily through residues located in the AA'BED β-sheets, which mediate the interactions between the C-terminal Ig domain in most Fcs[22–24,26,27]; the interface (including CHμ3, CHμ4 and Tp) is ~924 Å² and includes hydrogen bonds (Supplementary Fig. 3b, c).

Remarkably, despite sharing identical sequence and adopting Ig constant folds, the eight copies of CHμ4 exhibit structural differences that stabilize the tFcμ tetramer. Specifically, the C-terminal 38 residues (residues 411–448; FG loop, G strand and Tp) adopt one of two folds, with chains A and H adopting an A-type fold and chains B-G adopting a B-type fold (Fig. 2b, c). In the A-type fold, the FG loop and the G strand include residues 411–416 and 417–426, respectively, and the Tp folds back to form β-sheet interactions with the G strand (Fig. 2b). In the B-type fold, the FG loop and the G strand instead include residues 411–420 and 421–426, respectively, resulting in Tp β-strand (residues 434–441) being directed away from CHμ4 where they form β-sheet interactions with other Tp(s) rather than the G strand (Fig. 2c). Whereas chains A and H are superimposable, the six B-type domains adopt one of three distinct conformations based on the trajectory of the Tp and thus, each B-type domain is superimposable (RMSD of less than 0.18 Å) with one other B-type domain on the opposing side of the tetramer (Supplementary Fig. 4a). In sum, the A-type CHμ4 domains can be described as having a C-F-G-Tp sheet whereas B-type have a C-F-G sheet (with distinct F-G β-sheet interactions) and an extended Tp. Together these two folds facilitate the incorporation of each chain into the Tp assembly (discussed below).

### Interactions between Fcμs are mediated by CHμ4-CHμ4 and CHμ4-Tp interfaces

The tFcμ tetramer includes contacts between adjacent Fcμ monomers and Tps. Interactions between adjacent Fcμ monomers (e.g., Fcμ1-Fcμ2, Fcμ2-Fcμ3, and Fcμ3-Fcμ4) are dominated by a common set of residues in the CHμ4 CD loops, FG loops, and G-strands, forming an interface of ~1360 Å² (Fig. 3a). Notable residues mediating these contacts include CHμ4 G strand residues Met422, Thr424, and Arg423, which form electrostatic interactions and hydrogen bonds with G strand residues, CD loop (Asp359), and F strand (Ser406) residues in the neighboring

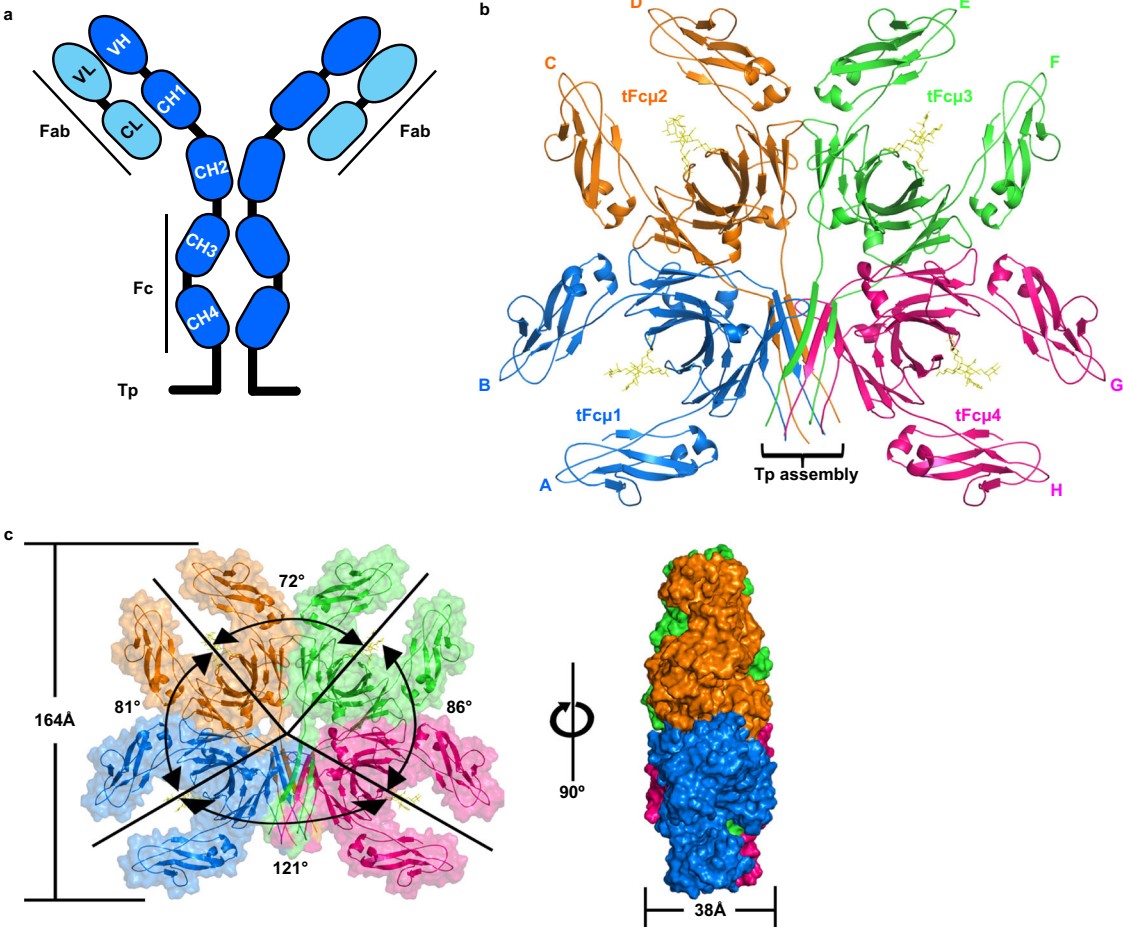

**Fig. 1 | Overall structure of teleost IgM-Fc core, tFcμ. a** Diagram of tIgM mono-mer. Constant (C) domains are represented as round corner rectangles and variable (V) domains are represented as ellipses. The heavy (H) chain is colored marine blue and the light (L) chain is colored light blue. **b** Cryo-EM structure of tFcμ (cartoon representation); chain IDs (A–H) and Fcμ1-4 and the Tp assembly are labeled. Gly-cans at residue Asn 374 are shown as yellow sticks. **c** Front view (*left*) and side view (*right*) of tFcμ tetramer, showing both cartoon representation and molecular sur-face. Values describing the angles between tFcμ monomers as well as the approx-imate diameter and thickness of the tFcμ tetramer are indicated. In panels **b** and **c**, tFcμ monomers are colored as indicated: *Blue*, tFcμ1. *Orange*, tFcμ2. *Green*, tFcμ3. *Pink*, tFcμ4.

CHμ4 domain. Additionally, FG loop residues Lys416 and Thr418 form interactions with CD loop Arg364 and F strand Tyr410, respectively (Fig. 3b, c).

Fcμ1 and Fcμ4 do not interface and instead interact with the Tp assembly, which is situated between the two Fcμs (Figs. 1b and 4a). Notably, Fcμ1 and Fcμ4 each contain one A-type CHμ4 domain and one B-type CHμ4 domain. The A-type domains neighbor the Tp assembly and contribute uniquely to its structure. Specifically, because A-type Tp β-strands form β-sheet interactions with CHμ4 G strands, residues in the pre-Tp loop (residues 427–431) are positioned to contact both the top and bottom Tp β-sheets, effectively clamping the Fcμ1 and Fcμ4 to the Tp assembly. Interactions between the pre-Tp loops and the rest of the Tp assembly are dominated by polar contacts between A-type pre-Tp loop residues and B-type Tp residues (Fig. 4d–f). Together Fc-Fc and Fc-Tp interactions form a band of contacts around the entire tetramer, with A-type CHμ4 domains contributing directly to the Tp assembly (Fig. 4d–f) and B-type CHμ4 domains forming interchain interactions between adjacent Fcμ monomers using CD loops, FG loops and G-strands (Fig. 3b, c).

### Tps from adjacent tFcμ monomers alternate to form two β-sheets that stabilize the Tp assembly

Within the tFcμ tetramer, β-strand pairing in the Tp assembly tightly links individual chains and also promotes stabilizing interactions between the two halves of the tetramer (Fcμ1-Fcμ2 and Fcμ3-Fcμ4). Specifically, we observe that Tp residues 434–440 form β-strands, which together constitute two 4-stranded β-sheets, each comprised of Tps from two adjacent tFcμ monomers (Fig. 4a). Notably, parallel Tp β-strand pairing alternates between Fcμs such that one sheet comprises Tps from chains A-C-B-D and the other comprises Tps from chains E-G-F-H; in other words, two Tp strands from the same Fcμ are separated by one Tp strand from an adjacent Fcμ (Fig. 4a). The two β-sheets are stacked, forming two planes rotated at an angle of ~48° relative to each other, and bury a hydrophobic core in-between. Buried residues include Val435, Leu437, and Leu439 from each chain, whereas hydro-philic residues Asn436, Ser438, and Asn440 face outwards (Fig. 4b, c).

Loop residues 441–448 follow the Tp β-strands. These residues are poorly ordered in the map; however, density suggests potential dis-ulfide bonds between copies of the residue Cys445 (Supplementary Fig. 4b–f). Residue 445 is a conserved Cys involved in interchain dis-ulfide bonding in mammalian IgA and IgM[23,24]; despite low local reso-lution, our map suggests possible disulfide bonding between Cys445[Fcμ1A] and Cys445[Fcμ3E], Cys445[Fcμ1B] and Cys445[Fcμ2C], Cys445[Fcμ2D] and Cys445[Fcμ4H], and Cys445[Fcμ3F] and Cys445[Fcμ4G] (see Supplementary Fig. 4b–f). The possibility of inter-chain disulfide bonding is further supported by higher molecular weight bands visible on non-reducing SDS-PAGE upon tFcμ purification (Supplementary Fig. 1b). Because potential Cys445 disulfide bonds are located in the C-terminal loop region following the

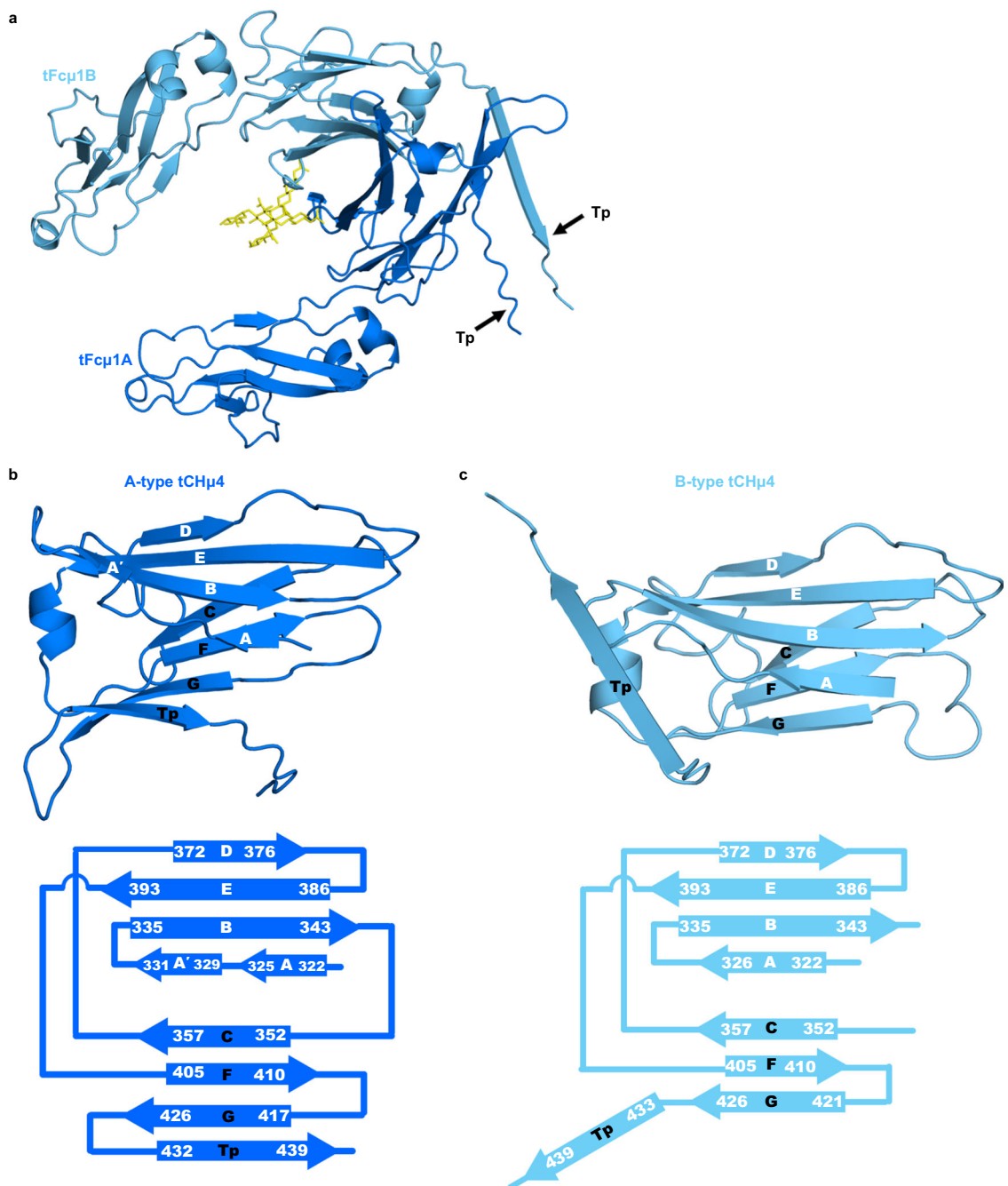

**Fig. 2 | tIgM CH4 domains adopt one of two Ig folds. a** Cartoon representation of tFcμ1 (chains A and B) with glycans shown in yellow; the orientation is the same as Fig. 1b. **b, c** A-type (panel **b**) and B-type (panel **c**) tCHμ4 folds. *Top*, cartoon representations of A-type tCHμ4 domain and B-type tCHμ4 domain. *Bottom*, topology diagram showing the secondary structure of tFcμ1A and tFcμ1B; terminal residues of each β-strand are numbered. B-type chains D and E contain A and A′ strands (not shown in this figure).

Tp β-strands, they do not appear to play a direct role in stabilizing the interactions between adjacent Fcs; nevertheless, they may effectively provide a covalent link between the two halves of the tetramer by connecting the top and bottom sheets of the Tp assembly (Fig. 4a and Supplementary Fig. 4b–f). Cysteine to serine mutation at this residue produced a tFcμ monomer in solution and eliminated evidence of interchain disulfide bonding, further suggesting a critical role in tIgM polymeric assembly (Supplementary Fig. 4g).

## Comparisons between teleost and human IgM
IgM is considered to be one of the evolutionarily oldest Igs[1]. It is omnipresent in all jawed vertebrates, except coelocanths where IgM has been lost, and teleost fish are the second oldest class expressing IgM, after cartilaginous fish[1,28]. To gain insights on pIg evolution we compared teleost and human (h) IgM structures[24,25]. Whereas tIgM is a tetramer, hIgM forms a JC-associated pentamer. Available human IgM structures contain the Fc region (hFcμ), JC (hJC), and human secretory component (hSC) (Fig. 5a). Overall, tFcμ and hFcμ adopt structures characterized by adjacent Fc monomers linked through Fcμ-Fcμ contacts and a Tp assembly; however, the structures are not globally superimposable and the relative orientation of the Fcμs, residues involved in Fcμ-Fcμ contacts, and the location and structure of the Tp assembly differ markedly, indicating that mechanisms of pIg assembly and function have evolved from teleost to humans.

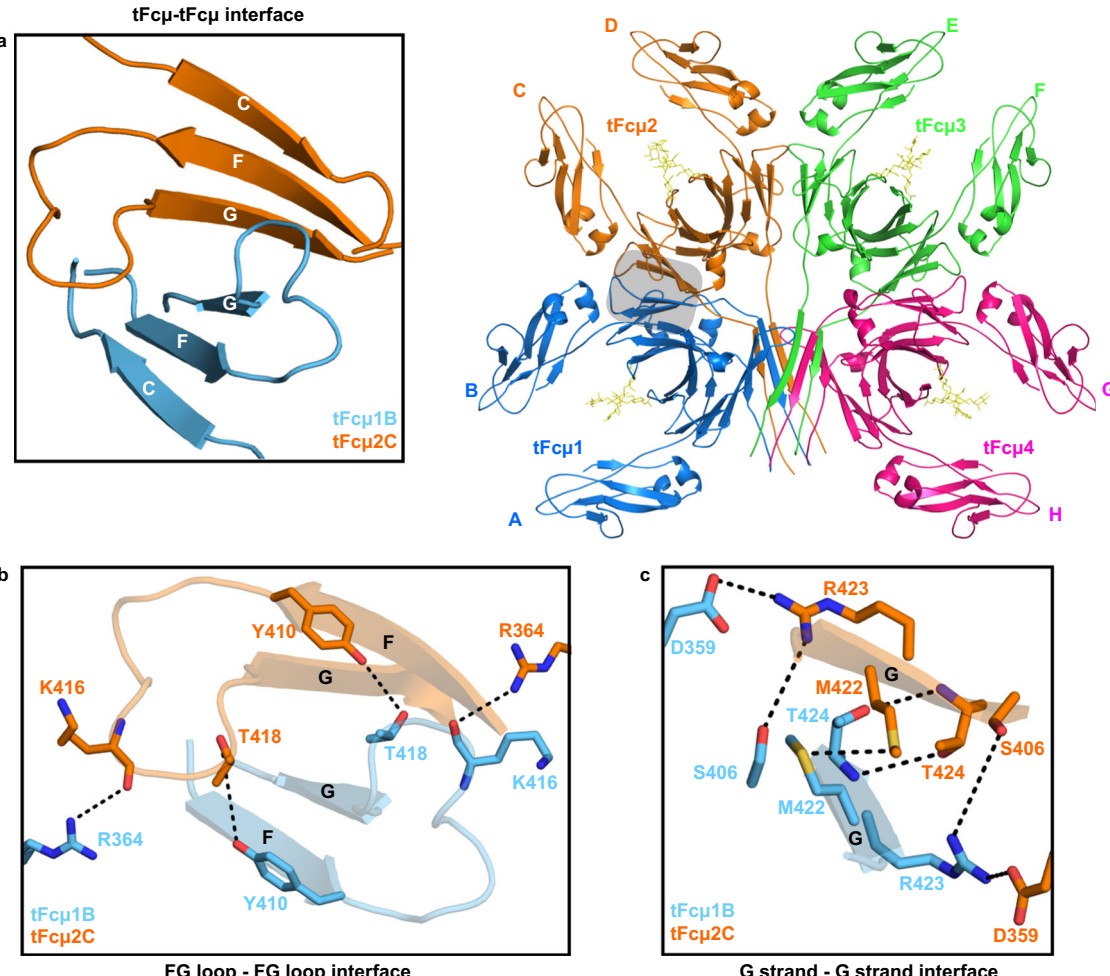

**Fig. 3 | The CHμ4 interface between adjacent tFcμ monomers. a** Overview of interactions between tFcμ1B and tFcμ2C at the CHμ4 interface (shaded area in tFcμ tetramer). *Left*, open-book overview of tFcμ-tFcμ interface boxed in the right panel, but shown in a different orientation. Only CHμ4 C-F-G β-sheets and FG loops are shown. *Right*, tFcμ tetramer with tFcμ1B and tFcμ2C at the CHμ4 interface in shaded box. **b** Hydrogen bonding mediated by FG loops between tFcμ1B and tFcμ2C at the CHμ4 interface. Cartoon representations are shown for the F strand, G strand, and FG loop. Residues participating in interactions are shown as sticks. **c** Electrostatic interactions and hydrogen bonding mediated by G strands between tFcμ1B and tFcμ2C at the CHμ4 interface. Cartoon representations of the G strands are shown, and interfacing residues are shown as sticks.

A fundamental difference between tFcμ and hFcμ is the geometric relationship among adjacent Fcμ monomers. For example, adjacent tFcμs are related by angles of 80–85° and ~120° (non-adjacent tFcμ1 and tFcμ4 separated by the Tp assembly) while the five adjacent hFcμs are related by angles between ~53° and ~62° (non-adjacent hFcμ1 and hFcμ5 separated by hJC) (Figs. 1b and 5a). The inter-Fc contacts that stabilize these geometries in both tFcμ and hFcμ include CHμ4 C-F-G β-sheets and FG loops; however, participating residues are not conserved, resulting in an interface of ~400Å² between hCHμ4s and a larger interface of ~800Å² between tCHμ4s. In addition, adjacent hFcμ monomer CH3 domains form contacts and are covalently linked by disulfide bonds between Cys414 residues[24,25] whereas this cysteine residue is not conserved in tFcμ and thus does not form inter-Fc disulfide bonds between tFcμ CH3 domains.

Structural comparisons between CH4 domains and Tps from human and teleost IgM are especially relevant to understanding pIg evolution and the functional differences between species because these domains form the majority of contacts that stabilize polymeric forms of IgM. Human and teleost IgM heavy chain constant domains share 28% sequence identity and 33% similarity among CH4 domains (Fig. 5b). A structural alignment between the human (chain B, excluding Tp residues) and teleost CHμ4 domains revealed a RMSD of 1.386 Å when aligned to A-type domains, and RMSD of 1.047 Å when

aligned to B-type domains, suggesting that B-type domains share greater structural similarity to human IgM CHμ4. Structural variability in the main chain arises primarily from differences in the length and conformations of the CD and DE loops (Fig. 5c), which contain residues that mediate contacts within and between Fcμs and are not well conserved from teleost to mammalian IgM. Even in regions that share high structurally similarity, such as the A-B-D-E β-sheet, some residues mediating contacts are not conserved, including those that stabilize the CH4-CH4 interface within each Fc (Fig. 5b).

The Tp has long been implicated in pIg assembly; it is presumed to be present in all pIgs[2,3] and is conserved among vertebrate pIg sequences, although residues near the C-terminus are variable, including the position of a conserved cysteine relative to the C-terminus (Supplementary Fig. 5). Mammalian IgM and IgA structures revealed Tps folding together with the JC to form β sandwich-like domains near the center of the complexes[22–24]. In human SIgM, residues 562–568 of each hFcμ Tp form a β-strand, and 10 copies form two parallel β-sheets that lie near-perpendicular to the central axis of the pentamer, positioning the distal β-strand from each sheet to pair with one β-strand from JC (Fig. 5a). In contrast, the tFcμ Tp assembly lacks a JC and is located toward the side of the complex between Fcμ1 and Fcμ4 with β-sheets positioned near-parallel to the central axis (Fig. 1b). In both structures, conserved hydrophobic residues Leu437 and

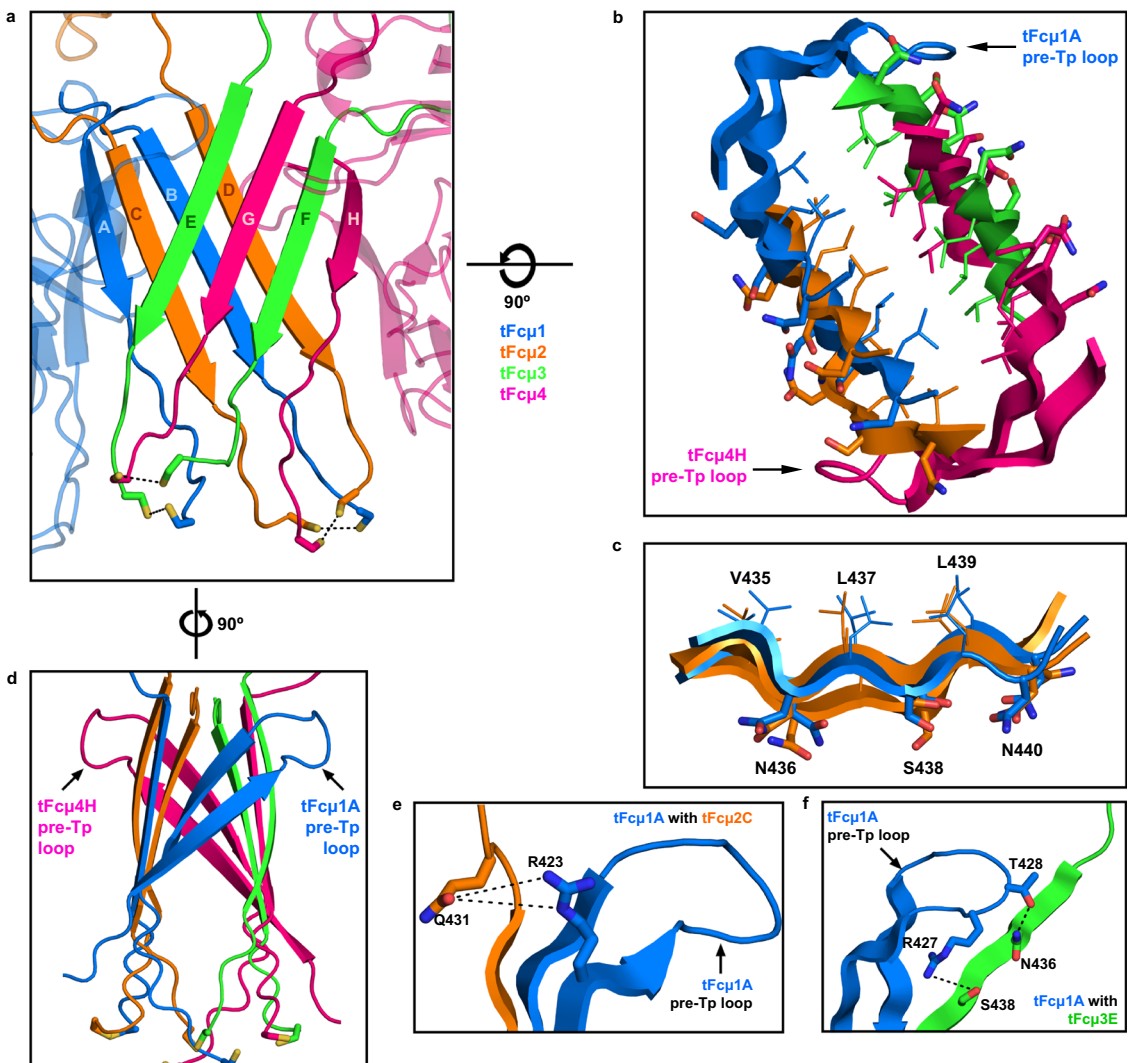

**Fig. 4 | Teleost IgM Tp assembly and its interactions with pre-Tp loop of A-type CHμ4. a** Cartoon representation of the tFcμ Tp assembly. The top β-sheet consists of Tp$^{tFcμ3E}$, Tp$^{tFcμ4G}$, Tp$^{tFcμ3F}$, and Tp$^{tFcμ4H}$ and bottom β-sheet consists of Tp$^{tFcμ1A}$, Tp$^{tFcμ2C}$, Tp$^{tFcμ1B}$, and Tp$^{tFcμ2D}$. Cys445 side chains are shown as sticks. Cys445 pairs that may form potential disulfides are linked using dashed lines. **b** Closeup view of the tFcμ Tp assembly hydrophobic core, which is depicted from the Tp C-termini looking toward the center of the tFcμ tetramer. **c** Closeup view of the tFcμ Tp assembly top β-sheet comprising β-strands from chains A-D. The side chains of hydrophobic residues are shown as lines while side chains of hydrophilic residues are shown as sticks. **d** Sideview of the tFcμ Tp assembly. **e** The tFcμ1A pre-Tp loop residues Arg423 interacts with Tp$^{tFcμ2C}$ residues Gln431; only chain A and chain C are shown. **f** The tFcμ1A pre-Tp loop residues Arg427 and Thr428 interact with Tp$^{tFcμ3E}$ residues Asn436 and Ser438; only chain A and chain E are shown.

Leu439 (Val564 and Leu566 in hFcμ) face the core of the β-sandwich and conserved hydrophilic residues Asn436 and Ser438 (Asn563 and Ser565 in hFcμ) face outward (Fig. 4b, c). However, despite the observation that conserved residues mediate hydrophobic interactions at the core of both hFcμ and tFcμ Tp assemblies, the relationship between the two Tp assembly β-sheets differs; the two tFcμ β-sheets are related by an angle of 48° whereas the two hFcμ β-sheets are related by an angle of 158° (Fig. 4a and Fig. 5a).

The most striking difference between tFcμ and hFcμ Tp assemblies is a difference in β-strand pairing. In tFcμ, each Tp pairs with a Tp originating from an adjacent Fc monomer while in hFcμ, Tp β-strands pair with a second Tp from the same hFcμ monomer. The exception to this is hFcμ3, which contributes one Tp to each β-sheet in the hFcμ Tp assembly (Figs. 4a and 5a). The hFcμ Tp strand-pairing pattern is also observed for Tps in polymeric forms of IgA[23]. Despite differences in strand pairing, the length from the last residue of CHμ4 (Asp426 in tIgM; Asp 553 in hIgM) to the first residue in the FcμTp β-strand (Leu434 in tIgM; Tyr562 in hIgM) is conserved between the tFcμ B-type

fold and hFcμ. Also conserved among hFcμ and tFcμ is a potential N-linked glycosylation site (PNGS) including residues Asn436 and Ser438. Earlier studies indicated that glycosylation at this site can influence hIgM polymerization and may prevent aggregation[24,29,30]. However, we did not observe ordered glycosylation at this site in the tFcμ structure (Supplementary Fig. 4h).

Finally, both tFcμ and hFcμ contain a conserved cysteine residue in the C-terminal loop following the Tp β-strand. While only four out of ten copies of Cys575 are ordered in published hFcμ structures[24], and density at this conserved cysteine (Cys445) is poor in the tFcμ structure (Supplementary Fig. 4b–f), data point toward an essential role for a cysteine near the C-terminus of IgMs. As noted, we observe putative disulfide bond formation linking tFcμ Cys445$^{tFcμ1A}$ with Cys445$^{tFcμ3E}$ and Cys445$^{tFcμ2D}$ with Cys445$^{tFcμ4H}$ and mutation of this residue inhibits tetramer formation (Supplementary Fig. 4g). In the case of hFcμ, this cysteine (Cys575) forms disulfides with the JC or with another Tp located in the same sheet and is reportedly necessary for hIgM assembly[24,25]. The pattern of

disulfide bonding appears to be different and to link different regions in tIgM versus hIgM. In sum, structural analysis of tFcμ and hFcμ reveals that despite conservation in sequence and length, IgM Tps can assemble into markedly different structures to promote assembly of pIgs.

## tFcμ motifs introduced into human heavy chains can promote the assembly of pIgs

Armed with additional structural insights and a desire to better understand pIg evolution and assembly mechanisms, we explored the ability of tFcμ motifs to facilitate the assembly of pIgs from other heavy

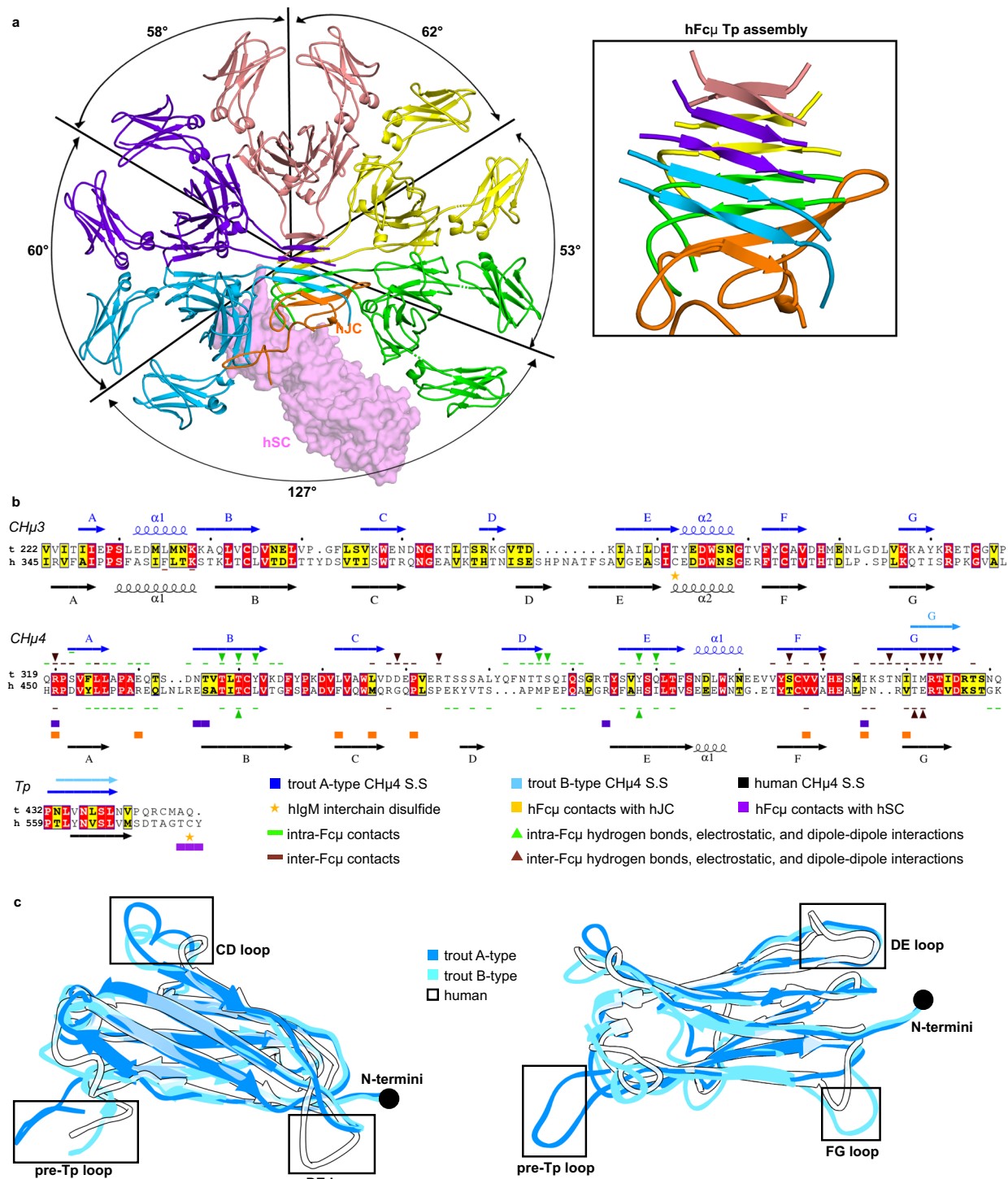

**Fig. 5 | Comparison of IgM structures from trout and humans. a** Human SFcμ structure, containing five hFcμ monomers, hJC, and hSC (PDB ID: 6KXS[24]); hFcμ and hJC are shown as cartoon representations and hSC is shown as molecular surface representation. Angles between Fcμ monomers are indicated in the structure. Inset, overview of hFcμ Tp assembly. Residues 560–576 of hFcμ heavy chain and part of hJC are shown and colored as in panel a. **b** Sequence alignment of IgM CHμ3, CHμ4, and Tp from *Oncorhynchus mykiss* (rainbow trout, t) and *Homo sapiens* (human, h), with secondary structure (S.S) and interchain contacts shown at top (tFcμ) and bottom (hSFcμ) of the sequences. For B-type tCHμ4, secondary structure features identical to A-type tCHμ4 are not indicated. In all panels, figures are annotated and colored according to the key. **c** Structural alignment of hFcμ CHμ4 domain with tFcμ A-type CHμ4 domain (chain A) and tFcμ B-type CHμ4 domain (chain B). All domains are shown as cartoon representations.

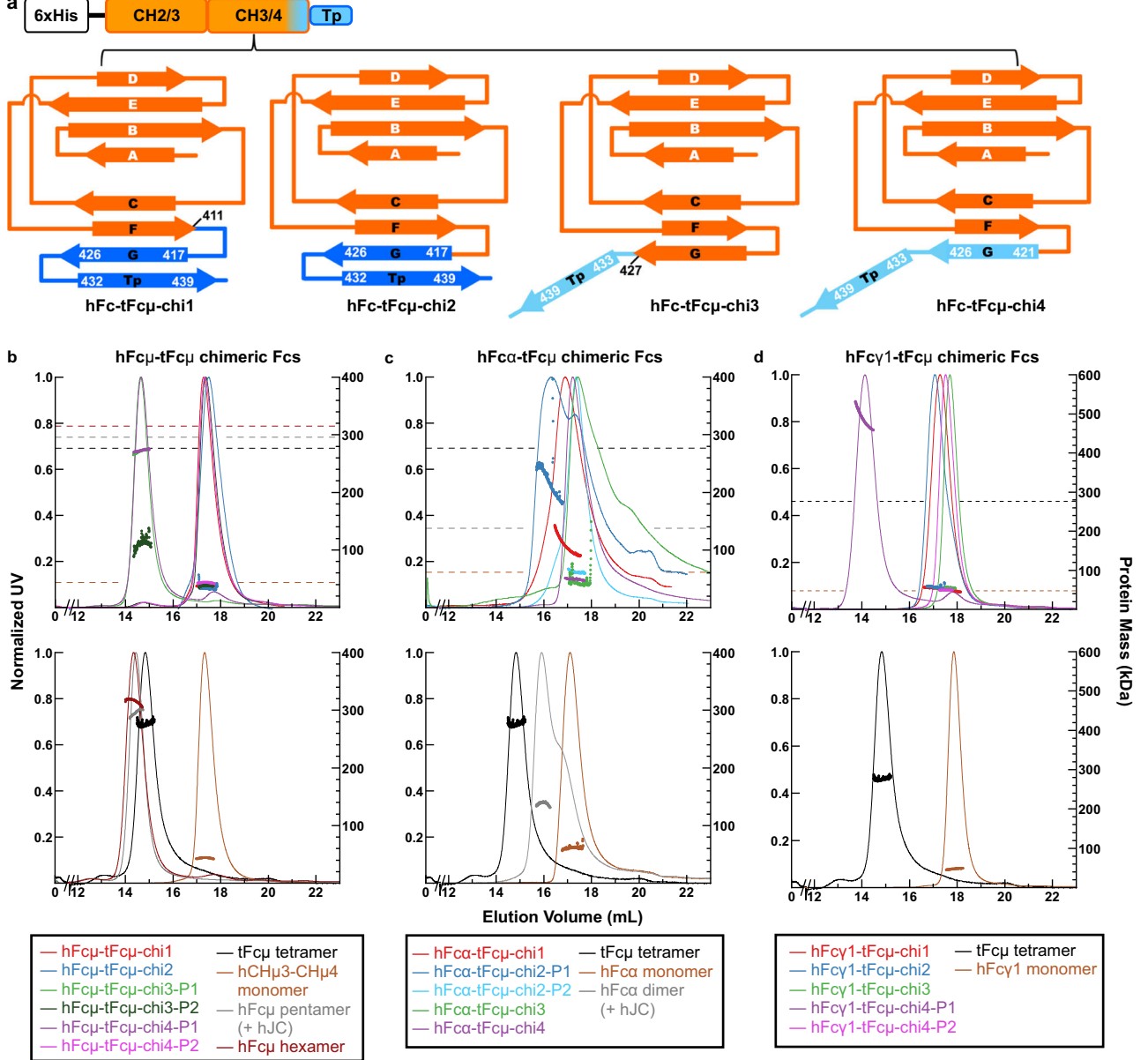

**Fig. 6 | Design and characterization of chimeric Fcs. a** *Top*, the design of chimeric Fc constructs; *bottom*, schematic showing topology of chimeric Fc constructs. Each chimeric Fc consists of an N-terminal hexa-histidine tag, partial hFc sequence (orange), and tFcμ C-terminal sequence (marine or light blue). **b–d** SEC-MALS chromatograms for the indicated chimeric Fcs (top) and associated control complexes (bottom). SEC elution profiles are shown as solid curves (left axis; normalized UV signal) and light scattering data, indicating protein mass, is shown as horizontal dots (right axis; protein mass in kDa). All data are colored according to the key below. In top panels, average protein mass of controls is indicated by horizontal dashed lines. **b** hFcμ-tFcμ chimeric Fcs and relevant controls. **c** hFcα-tFcμ chimeric Fcs and relevant controls. **d** hFcγ1-tFcμ chimeric Fcs and relevant controls. For all samples, the predicted molecular weight from the protein sequence, the average molecular weight determined by MALS, and the number and potential of PNGS are listed in Supplementary Table 4. The amino acid sequences of chimeric Fc used in this study are provided in Supplementary Table 5. Source data are provided as a Source Data file.

chains including human mu (IgM), alpha-1 (IgA1) and gamma-1 (IgG1). The tFcμ structure suggests that both A-type and B-type tCHμ4 domains are essential for tFcμ assembly. Given that tFcμ A-type and B-type tCHμ4 domains, despite having the same sequence, differ structurally in the FG loop, G strand, and pre-Tp loop regions, we hypothesized that the mechanisms of tFcμ assembly rely heavily on residues in these motifs. To test this hypothesis, we designed twelve chimeric expression constructs, in which residues found in the A-type or B-type tCHμ4 G strand and/or Tp replaced endogenous counterparts in human Fcμ, Fcα1, and Fcγ1 (Fig. 6a). Chimeric Fc constructs and associated controls were transiently transfected into human cells and the resulting protein complexes were purified and their polymeric

state(s) evaluated using size exclusion elution chromatography with in-line multi-angle light scattering (SEC-MALS), which reports on the absolute molecular weight of target proteins and complexes[31].

As noted, human IgM is naturally expressed as a hexamer or a JC-containing pentamer, and the Tp is reportedly essential for the assembly of both complexes[30]. Consistent with published studies, in our experimental system the removal of Tp residues resulted in only monomeric hFcμ (Fig. 6b and Supplementary Fig. 6a). To determine if the tFcμ Tp could promote pIg assembly in hFcμ constructs lacking the endogenous Tp (and the JC), we measured the molecular weight of four chimeric proteins, hFcμ-tFcμ-chi1 to chi4 (Fig. 6b and Supplementary Fig. 6a). The hFcμ-tFcμ-chi1 and hFcμ-tFcμ-chi2 were

monomeric; however, hFcμ-tFcμ-chi3 and hFcμ-tFcμ-chi4 were polymeric with average molecular weights consistent with tetrameric complexes (Fig. 6b and Supplementary Table 4). The hFcμ-tFcμ-chi3 contains the tFcμ Tp and hFcμ-tFcμ-chi4 contains the B-type tCH4 G strand and tFcμ Tp residues; thus, data indicate that substitution of hFcμ residues with these motifs, but not the A-type tCH4 G strand residues linked to the tFcμ Tp, can promote pIg assembly. However, in contrast to the hexamers observed when full-length hFcμ is expressed without JC, chimeric pIgs adopted probable tetramers.

We also sought to determine if the tCH4 G strand and/or Tp residues could promote pIg assembly in the context of hFcα, which is monomeric in the absence of the JC and typically dimeric when expressed with a JC, or in the context of Fcγ1, which is naturally monomeric and does not contain a Tp. In contrast to hFcμ-tFcμ chimeras, we found that hFcα-tFcμ-chi1 and hFcα-tFcμ-chi2 formed pIgs whereas hFcα-tFcμ-chi3 and hFcα-tFcμ-chi4 did not. The molecular weight of resulting pIgs was consistent with a mix of species ranging from dimers to tetramers (Fig. 6c and Supplementary Fig. 6b). Data signify that in the context of the human alpha-1 heavy chain, the G strand and Tp residues found in A-type tCH4 fold can promote pIg assembly in the absence of JC, whereas G strand and Tp residues found in the B-type tCH4 fold, or Tp residues alone, cannot. Among hFcγ1-tFcμ chimeras, only hFcγ1-tFcμ-chi4 promoted pIg assembly, suggesting that in the context of the human gamma-1 heavy chain, substituting G strand and Tp residues found in the B-type tCH4 fold could promote assembly whereas others could not. For all chimeric Fc constructs that promoted pIg assembly, pIgs were isolated from mixtures containing mIgs before SEC-MALS analysis (Supplementary Fig. 6). Among hFcμ-tFcμ chimeras, the pIgs represented more than 50% of the total protein purified whereas pIgs isolated from hFcα-tFcμ and hFcγ1-tFcμ chimeras represented less than 30% of the total protein purified, with mIgs making up the rest of the total (Supplementary Fig. 6). This suggests that in our experimental system, pIg assembly promoted by tFcμ motifs was most efficient for hFcμ-tFcμ chimeras.

## Discussion

Throughout evolution, pIgs have evolved critical, yet incompletely characterized functions in jawed vertebrates. Data reported here, together with recently published mammalian pIg structures, demonstrate that pIgs are a structurally diverse group of antibodies varying in composition, size, and conformation. For the host, assembly of these complexes is a critical first step to implementing a vast array of downstream effector functions and we find evidence that across species, this can be accomplished through several distinct yet overlapping mechanisms. Specifically, the Ig domains and Tps provide a common set of building blocks, while differences in heavy chain amino acid sequence and the presence or absence of JC are variables adding plasticity to the process.

As noted, most jawed vertebrates, including mammals, birds, amphibians, reptiles, and cartilaginous fish, reportedly encode JC and can incorporate it into pIgM, whereas teleost fish have lost JC and assemble the Fc regions of pIgs using only Ig domains and Tps[4–6]. We find that this unique teleost characteristic is associated with distinct tFcμ structural folding patterns when compared to mammalian pIgM structures containing the JC[24,25]. This raises the possibility that JC-independent, and JC-dependent assembly evolved to employ distinct biophysical mechanisms. However, we also find that tFcμ C-terminal residues (and presumably associated structural motifs) are able to promote JC-independent pIg assembly of otherwise human heavy chains, suggesting some degree of interchangeability among the sequences of pIg-forming heavy chains and assembly mechanisms. For example, chimeric hFcμ-tFcμ revealed that a subset of tFcμ motifs were able to promote pIg assembly of an otherwise human IgM heavy chain in the absence of JC. It is intriguing that this effect was found only when the tFcμ CH4 G strand and/or Tp residues found in the B-type fold, but

not the A-type fold, were used. While this may reflect the higher degree of similarity shared between the B-type fold and human IgM domains, the resulting pIgs appeared to be tetramers rather than hexamers, the predominant polymeric state associated with human IgM lacking the JC in our experiments (and also in human serum)[10]. It remains unknown whether hFcμ-tFcμ chimera tetramers assemble into structures more similar to tIgM or hIgM with respect to the Tp assembly structure, contacts, and geometric relationships between the incorporated Fcs. While pIg assembly of hFcα-tFcμ, and hFcγ1-tFcμ chimeras was notably less efficient than hFcμ-tFcμ chimeras, it is significant that tFcμ components could promote some pIg assembly in the context of human IgA1 and IgG1 heavy chains, neither of which naturally form pIgs in the absence of JC. While we cannot rule out the possibility that assembly in plasma cells differs from the HEK cells used in our experiments, experimental cell lines assembled controls of expected MW and provided a uniform system to evaluate how sequence differences among chimeras influences pIg assembly.

Despite the prevalence of JC in the genomes of most jawed vertebrates (other than teleosts), some species assemble pIgs in the absence of JC. For example, human IgM can assemble without JC (typically hexameric) and amphibian IgX (polymeric) also appears to lack the JC[5,6,10–12]. This raises the question of whether or not pIgs lacking the JC from species other than teleost fish have structures most similar to hexameric hIgM or to tetrameric tIgM. At this time, a high-resolution structure detailing the Tp assembly of hexameric IgM has not been reported; however, negative-stain EM and small-angle light scattering (SAXS) data suggest all six hFcμs form Fc-Fc contacts, and that the Tp assembly remains in the center of the complex as seen in pentameric IgM, rather than bridging two Fcs as we observed in tetrameric tFcμ[32,33]. The structures of unliganded Tps (e.g., those found in monomeric IgA or IgM) are also unknown and therefore it remains unclear if and how Tps might change conformation during mammalian pIg assembly in absence of JC and/or if they adopt conformations distinct from what is observed in mammalian pIg and SIg structures. Without such information, it is difficult to judge whether the Tp β-strands would alternate in hIgM hexamers, as observed in tFcμ tetramers, or if they would pair with another Tp β-strand from the same Fc as observed in hFcμ pentamer and mammalian pIgA.

While our structural data provide considerable insight on tIgM structure and assembly mechanisms, the stages of the assembly process remain unclear. Identification of two distinct tCH4 folds evokes the question of how tIgM heavy chains fold and if they adopt A- and/or B-type CH4 domains prior to the assembly of pIgM. In isolation, the A-type fold appears likely to be more stable than the B-type fold because the β-strand region of the Tp forms β-sheet interactions with the G strand, rather than being extended away from the core of the Ig domain in a potentially flexible location (see Fig. 2b, c). While further study is needed to define the stages of tFcμ assembly, one possibility is that heavy chains first adopt the A-type conformation and that polymeric IgM assembly is triggered when two tFcμ monomers come into proximity and interactions at the Fc-Fc interface displace Tp β-strands from both Fcs, effectively kicking them out of the Ig-fold and allowing them to form parallel β-sheet interactions with each other. It is unclear if there is sufficient energy in the tFcμ-tFcμ interface for its formation to break Tp-G strand hydrogen bonding and thus if this model is correct, it is conceivable that molecular chaperones, which have been implicated in mammalian pIgA and pIgM assembly processes, are involved[34,35]. Regardless, sequential interactions between Fcs could add strands to the Tp assembly until the tetramer was closed by the last remaining A-type fold. In a related model, sequential addition of heavy chains might culminate in dimers (tetramer halves), two of which subsequently come together to form a tetramer. These models are consistent with the alternating Tp β-strand pairing that we observe in tFcμ, but that is not seen in mammalian IgM or IgA containing JC[22–24]. Furthermore, given that Cys445Ser mutation abolished tFcμ tetramer

formation (Supplementary Fig. 4g), it is possible that disulfide bonds linking non-adjacent Fcs, and/or the top and bottom β-sheets in the Tp assembly (see Fig. 4a, Supplementary Fig. 4b–f) stabilize some part of this process. A critical role for Cys445 is consistent with published work indicating that disulfide crosslinking within tIgM influences antibody assembly, stability, and half-life[17,36,37]; however additional studies will be needed to work out its precise role(s) in these processes.

Another remarkable finding of our work is that the majority of pIg assembly mechanisms appear likely to culminate in the release of an asymmetric pIg complex, despite being assembled from multiple copies of identical Ig-heavy chains with identical sequence. Asymmetry in mammalian pIgM and pIgA was verified with the publication of pIgA and pIgM structures, in which the JC was bound to two Igs uniquely, and is thus postulated to template the addition of monomers during assembly and to confer asymmetry on the resulting complex[22–24]. However, the tFcµ structure, which lacks a JC, also reveals elements of asymmetry, with four Fc monomers directed away from the Tp assembly such that the relative geometric relationships between each tFcµ are not identical. As noted, this arrangement appears to be dependent upon the existence of A-type and B-type CHµ4 domains, yet it is unclear if asymmetry in pIgs evolved as an efficient means to assemble a pIg and/or if asymmetry provides a functional advantage.

IgM is a critical Ig heavy chain class found in all jawed vertebrates. Studies in mammals demonstrate that its effector functions are mediated by high avidity interactions with antigen (from five sets of Fabs), complement activation, binding to IgM-specific Fc receptors (FcRs), and mucosal functions associated with pIgR-dependent secretion. While relatively few reports have documented its function in teleost fish, evidence suggests that tIgM also participates in complement activation, mucosal functions such as bacterial coating and immune exclusion, and binds IgM-specific FcRs[21,36,38,39]. The tetrameric state of tIgM we observe implies that it has the potential to bind multiple copies of an epitope simultaneously, which is likely to support high-avidity antigen binding and/or antigen crosslinking. The structural relationship between tFcµ CH2 and Fab domains remains uncharacterized and thus modeling potential positions of tIgM Fabs is non-trivial; however, based on recent structures detailing the hIgM CH2 domain we estimate they might extend ~125 Å from the center of the complex[40]. Furthermore, our structure indicates that Fabs are likely to be directed away from the Tp assembly, which separates Fcµ1 and Fcµ4, leaving portions of the Tp assembly and its adjacent Fcs sterically accessible. It is functionally intriguing that each copy of tFcµ is related to another by a different angle, raising the possibility that different geometric relationships between Fabs within a pIg provide unique binding advantages for different types of antigen distribution. For example, Fcs separated by 120° might bind epitopes that are separated by larger distances compared to Fcs separated by 80° (Fig. 1c). We also hypothesize that the 120° separation between tFcµ1 and tFcµ4 could promote FcR binding to tFcµ1 and/or tFcµ4. We note that the two Fcs in dimeric forms of mammalian IgA are related by angles of 97°; in human IgM, two Fcs are separated by 53°–62°, with exception of Fcµ1 and Fcµ5 which are separated by an angle of 127° (see Fig. 5a). The 97° bend between two IgA monomers in the SIgA structure is predicted to increase accessibility to FcR binding sites and promote binding away from Fabs[22]. Future studies will be necessary to determine if the geometric relationship between Ig monomers in pIgs have been selected for because of a functional advantage, an assembly advantage, or both.

Although tSIgT has been identified as the immunoglobulin specialized in teleost mucosal immunity, substantial amounts of tSIgM are also found in gut and skin mucus before and after immunization[18,21,41]. SIgs are crucial to mucosal immunity because of their ability to bind and exclude microbes before they breach epithelial cells. This is especially important to fish since their skin is covered in mucus and constantly exposed to an external aquatic environment. In both mammals and teleosts, the pIgR binds and transports pIgM across epithelial cells and is cleaved from the cell surface upon delivery to the mucosa, leaving its ectodomain bound to the pIg where it is referred to as secretory component (SC). In mammals, the JC is necessary for this process and forms direct contacts with SC, which contains five Ig-like domains[24,25]. In contrast, JC is not required for delivery of tSIgM to the mucosa and tSC contains just two Ig-like domains that share low sequence conservation with mammalian counterparts; these observations indicate an alternative tSC-tIgM binding mechanism[19,42]. Consistent with these observations, we found that only two hIgM residues that bind hSC are conserved and solvent-accessible on tFcµ (Fig. 5b). It is yet unknown where the tSC binding site is located and if that site overlaps with other FcR binding sites.

## Methods

### Construct design and protein expression

The gene encoding the *Oncorhynchus mykiss* IgM heavy chain secretory form (GenBank: AAW66974.1) was obtained from the NCBI database (https://www.ncbi.nlm.nih.gov/). DNA sequence encoding a signal peptide (MFPASLLLLLAAASCVHC), a hexa-histidine tag, CHµ3 (residues 222–318), CHµ4 (residues 319–432), and Tp (residues 433–448) was codon optimized for human cell expression, synthesized (Integrated DNA Technologies, Inc.), and cloned into mammalian transient expression vector pD2610-v1 (Atum). Resulting plasmid DNA was transiently transfected into HEK Expi293-F cell line (Gibco, A14528) using the ExpiFectamine 293 Transfection Kit (Gibco). Six days after transfection, supernatants were harvested and His-tagged tIgM-Fc (tFcµ) protein was purified using Ni-NTA Agarose (Qiagen) and subsequently, Superose 6 Increase 10/300 column (Cytiva) size exclusion chromatography (SEC). Elution fractions corresponding to the predicted molecular weight of tFcµ tetramer were collected and stored in TBS buffer (containing 20 mM Tris-HCl and 150 mL NaCl, pH 7.8) for cryo-EM.

To design chimeric Fcs, published structures containing hFcα, hFcγ1, and hFcµ (PDB IDs: 6UE7[23], 1MCO[43], 6KXS[24], respectively) were structurally aligned to A-type and B-type tCHµ4 structures. Constructs hFc-tFcµ-chi1 and hFc-tFcµ-chi2 were designed based on structural alignment to A-type tCHµ4 and constructs hFc-tFcµ-chi3 and hFc-tFcµ-chi4 were designed based on structural alignment to B-type tCHµ4. In construct hFc-tFcµ-chi1, A-type tCHµ4 residues in FG loop, G strand, pre-Tp loop, and Tp (residues 412–448) were swapped into the C-terminal CH domain of hFcs (CH3 or CH4). In the hFc-tFcµ-chi2 construct, A-type tCHµ4 residues in G strand, pre-Tp loop, and Tp (residues 417–448) were swapped into the C-terminal CH domain of hFcs (CH3 or CH4). In the hFc-tFcµ-chi3 construct, B-type tCHµ4 residues in the pre-Tp loop, and Tp (residues 427–448) were swapped into the C-terminal CH domain of hFcs (CH3 or CH4). In the hFc-tFcµ-chi4 construct, B-type tCHµ4 residues in G strand, pre-Tp loop, and Tp (residues 421–448) were swapped into the C-terminal CH domain of hFcα and hFcµ, or fused after the C-terminus of hFcγ1. Chimeric Fc constructs were cloned, transfected, and purified using the same protocol described for the tFcµ construct.

### SEC-multiangle light scattering (SEC-MALS) analysis

Proteins samples were prepared as described. After purification by SEC (see Supplementary Figs. 1 and 6), fractions in each peak were pooled together, concentrated to approximately 2 g/L, filtered, and sent to University of Pennsylvania for SEC-MALS analysis. SEC-MALS experiments were performed on tFcµ and mutant variants. 100uL injections of purified protein in TBS buffer was loaded onto a Superose 6 Increase 10/300 column (Cytiva) at 0.5 ml/min at 25 °C. Absolute molecular weights were determined using MALS. The scattered light intensity of the column eluant was recorded at 18 different angles using a DAWN-HELEOS MALS detector (Wyatt Technology Corp.) operating at 658 nm after calibration with the monomer fraction of Type V BSA (Sigma).

The protein concentration of the eluant was determined using an in-line Optilab T-rex interferometric refractometer (Wyatt Technology Corp.). The weight-averaged molecular weight of species within defined chromatographic peaks was calculated using the ASTRA software version 8.0 (Wyatt Technology Corp.), by construction of Debye plots (KC/$R\theta$ *versus* $sin^2[\theta/2]$) at 1-s data intervals. The weight-averaged molecular weight was then calculated at each point of the chromatographic trace from the Debye plot intercept, and an overall average molecular weight was calculated by averaging across the peak. Data were further analyzed using the protein conjugate analysis module as implemented in Astra for protein with carbohydrate modifier. For all samples, the predicted molecular weight from the protein sequence, the average molecular weight determined by MALS, and the number and potential of PNGS are listed in Supplementary Table 4[44].

### Cryo-EM grid preparation and data collection
UltrAuFoil R1.2/1.3 300 mesh grids (Quantifoil) were glow discharged in a Pelco easiGlow (TedPella Inc) for 90 seconds at 25 mA current. Using a Vitrobot Mark IV (Thermo Fisher) grids were prepared at 4 °C, 100% RH, blot force of 5, 0 s wait and drain time, and with blot time ranging from 2 to 8 second. Sample was diluted from 1.8 mg/mL to 0.1 mg/mL prior to freezing. Movies were collected using SerialEM on a Titan Krios G3 cryoTEM (Thermo Fisher) operating at 300 kV, equipped with BioQuantum Energy Filter (20 eV slit width, Gatan) and a K3 direct electron detector (Gatan) using beam image-shift in a 3x3x3 pattern. A total of 3618 movies were collected at 105,000 magnification in super-resolution mode with the corrected pixel size of 0.427 Å/pix, ~60 electros/Å² total dose, and 40 frames per movie.

### Cryo-EM data processing
Movies were imported into cryoSPARC and binned by 2 during patch motion correction[45,46]. Images were curated for CTF fits, ice thickness, and contamination resulting in 3131 remaining images. Blob picker was used with about half of the images to generate an initial particle stack. After 2D classification, best classes were selected as templates for the Template picker on the full dataset, resulting in ~1.3 M picks. Particles were binned by 2 during extraction and subject to several rounds of 2D classification and heterogeneous refinement. The remaining 698k particles were re-extracted at the full pixel size. After an additional round of heterogeneous refinement, 610k particles were used in non-uniform refinement (with C2 symmetry), resulting in a 2.9 Å map[47]. After a cycle of local and global CTF refinement, an improved map at 2.78 Å was obtained. A non-uniform refinement with C1 symmetry was also computed, resulting in a 2.9 Å map. C1 and C2 maps were essentially superimposable and the C2 map was used for structure refinement.

### Atomic model building, refinement, and validation
A homology model for a single tCHμ4 domain was made with SWISS-MODEL using the human IgM-Fc core cryo-EM structure (PDB ID: 6KXS[24]) as template; the C-terminal Tp was not included in the homology model[48]. The tCHμ4 homology model was docked into real-space density of CHμ4$^{Fc\mu1A}$ using UCSF Chimera, developed by the Resource for Biocomputing, Visualization, and Informatics at the University of California, San Francisco, with support from NIH P41-GM103311[49]. Initial inspection of the docked homology model revealed several regions poorly fitted into density. To improved model fitting, less structured regions (residues Val362 to Gln378, C-terminal sequence after residue Arg427) were deleted from the homology model and resulting model was automatically fitted into real-space density using Coot Molecular Graphics Package[50]. Subsequently, residues 397–427 were deleted from the model due to mis-fitting and then rebuilt manually. The outcome was inspected and manually rebuilt in Coot before being subjected to Phenix real-space refinement[51]. After several refinement iterations, the model of CHμ4$^{Fc\mu1A}$ and its Tp agreed well with the cryo-EM density map and a copy of the model was docked into density of CHμ4$^{Fc\mu1B}$. Due to the structural differences between CHμ4$^{Fc\mu1A}$ and CHμ4$^{Fc\mu1B}$, adjustments to the C-terminal sequence of CHμ4$^{Fc\mu1B}$ were made manually in Coot to optimize model fitting into cryo-EM map. Models of CHμ4$^{Fc\mu1A}$ and CHμ4$^{Fc\mu1B}$ were combined and refined using Phenix and Coot Molecular Graphics Package[50,51]. Copies of CHμ4$^{Fc\mu1B}$ were docked into the density of CHμ4$^{Fc\mu2C-4G}$ and one copy of CHμ4$^{Fc\mu1A}$ was docked into the density of CHμ4$^{Fc\mu4H}$ using UCSF Chimera; these newly docked CHμ4 models were first refined individually using Phenix and then combined with previously refined CHμ4$^{Fc\mu1A}$ and CHμ4$^{Fc\mu1B}$ domains[49,51]. Residues linking CHμ4 domains to Tps and residues in the Tp assembly were built manually into the combined model. Cys445 residues were built into density with rotamers that assumed the preferred orientation to form interchain disulfides. The resulting model was subjected to iterations of automatic refinement in Phenix and manual adjustment in the Coot Molecular Graphics Package. The refined CHμ4 model was used as template to make a homology model for the tCHμ3 domain using SWISS-MODEL[48,50,51]. The CHμ3 homology model was docked into the density of CHμ3$^{Fc\mu2D}$ and refined using similar methods as described above. Subsequently, copies of CHμ3$^{Fc\mu2D}$ were docked into the density of remaining CHμ3 domains. CHμ3 domains were joined with the previously refined CHμ4 domains and Tp assembly; the entire model is refined using Phenix and Coot Molecular Graphics Package[50,51]. The final tFcμ structure was evaluated by Phenix EM Validation, MolProbity, and EMRinger[51,52]; results are summarized in Supplementary Table 1.

### Sequence and structural alignments and structural analysis
NCBI Genbank accession numbers for Ig heavy chain sequences shown in figures are: *Oncorhynchus mykiss* (trout) IgM secretory form (AAW66974.1), *Salmo Salar* (salmon) IgM (AAB24064.1), *Xenopus laevis* (frog) IgM (AAA49774.1), *Trachemys scripta elegans* (turtle) IgM (AFR90255.1), *Gallus gallus* (chicken) IgM (P01875), *Mus musculus* (mouse) IgM (P01872), and *Homo sapiens* (human) IgM (P01871). Sequences alignments were made using ClustalOmega and corresponding figures were made using Espript3[53,54]. Structural alignments between human IgM CH4 of (residues 450–555 of chain A; PDB ID: 6KXS[24]) and teleost IgM-CH4 (residues 316–428 of chain A or chain B) were made using the "align" function in PyMOL; structural alignments between teleost IgM CH3-CH4-Tp (residues 225–445) were made using the "align" function in PyMOL[55]. Interfaces were calculated using PISA, "Protein Interfaces, surfaces and assemblies" service at the European Bioinformatics Institute[56]; results are summarized in Supplementary Table 2.

### Fc angle and distance measurements
To measure the radius of polymeric forms of Fcμ, the tFcμ (this study) and human SIgM-Fc core (PDB ID: 6KXS[24]) structures were imported into UCSF Chimera[49]. For each model, a plane was defined using built-in "structure measurements" tools and the radius was calculated automatically. To measure angles between adjacent Fcs, Tp sequences were removed from models. An axis was defined using mass weighting and helical correction for each Fc monomer and then angles between the defined axes were measured and defined as the angles between adjacent Fcs. To measure the diameter of the solvent-accessible hole at the center of tFcμ tetramer, tFcμ model was imported into UCSF Chimera[49]. Nine atom pairs located on the surface were chosen around the hole. The distance between two atoms in each pair was measured and the diameter of this hole was determined to be the average of these distances. Chosen atom pairs and the distance measurements from each pair are summarized in Supplementary Table 3.

To measure the angle between the two tFcμ-Tp β-sheets, the tFcμ tetramer structure was imported into PyMOL[55]. The angle between Tp$^{tFcμ1A}$ and Tp$^{tFcμ3E}$ was measured as the dihedral angle between Cα atoms of Leu434$^{tFcμ1A}$, Asn440$^{tFcμ1A}$, Asn440$^{tFcμ3E}$, and Leu434$^{tFcμ3E}$; angles between Tp$^{tFcμ1B}$ and Tp$^{tFcμ3F}$, Tp$^{tFcμ2C}$ and Tp$^{tFcμ4G}$, Tp$^{tFcμ2D}$ and Tp$^{tFcμ4H}$ were measured using the same method. The angle between two Tp β-sheets was determined to be the average of these measurements. To measure the angle between the two hFcμ-Tp β-sheets the hSIgM-Fc core structure (PDB ID: 6KXS[24]) was imported into PyMOL[55]. The angle between Tp$^{Fcμ1A}$ and Tp$^{Fcμ5L}$ was measured as the dihedral angle between Cα atoms of Tyr562$^{Fcμ1A}$, Val567$^{Fcμ1A}$, Val567$^{Fcμ5L}$, and Tyr562$^{Fcμ5L}$; angles between Tp$^{Fcμ1B}$ and Tp$^{Fcμ5K}$, Tp$^{Fcμ2C}$ and Tp$^{Fcμ4H}$, Tp$^{Fcμ2D}$ and Tp$^{Fcμ4G}$, Tp$^{Fcμ3E}$ and Tp$^{Fcμ3F}$ were measured using the same method. An average of these measurements was taken and the angle between two Tp β-sheets was determined to be the supplement angle of the average.

To measure the angle between tFcμ CH3 and CH4, the tFcμ tetramer model was imported into PyMOL[55]. For each copy of heavy chain, the angle between tFcμ CH3 and tFcμ CH4 was measured as the dihedral angle between the Cα atoms of His298, Lys238, Glu412, and Thr332. The average of eight angle measurements was reported as the final value of tFcμ CH3-CH4 angle.

## Figures
Structural figures were made using UCSF Chimera and PyMOL[49,55]. SEC data and SEC-MALS data were replotted using Origin. All figures were assembled using Adobe Photoshop and Adobe Illustrator.

## Reporting summary
Further information on research design is available in the Nature Portfolio Reporting Summary linked to this article.

## Data availability
The tFcμ tetramer Cryo-EM data generated in this study have been deposited in the EM databank (www.ebi.ac.uk/emdb) with the accession code EMD-40054. The refined coordinates generated in this study have been deposited in the Protein Data Bank (www.rcsb.org) with accession code 8GHZ. Previously published protein structure data used for analysis in this study are available in the Protein Data Bank (www.rcsb.org) under PDB ID: 6KXS[24] (human hSIgM-Fc core), 6UE7[23] (human hSIgA-Fc core), 1MCO[43] (human IgG1-Fc). Source data are provided with this paper. The protein sequences used for sequence alignment are available in NCBI database (https://www.ncbi.nlm.nih.gov/) under these accession codes: *Oncorhynchus mykiss* (trout) IgM secretory form (AAW66974.1), *Salmo Salar* (salmon) IgM (AAB24064.1), *Xenopus laevis* (frog) IgM (AAA49774.1), *Trachemys scripta elegans* (turtle) IgM (AFR90255.1), *Gallus gallus* (chicken) IgM (P01875), *Mus musculus* (mouse) IgM (P01872), and *Homo sapiens* (human) IgM (P01871). All other data are available in the article and its Supplementary files or from the corresponding author upon request. Source data are provided with this paper.

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

## Acknowledgements

We thank Kushol Gupta (University of Pennsylvania) for assistance with SEC-MALS data collection and analysis and thank members of the Stadtmueller Laboratory for insightful conversations and suggestions related to this work. (Cryo) Electron microscopy was performed in the Beckman Institute Resource Center for Transmission Electron Microscopy at Caltech. This work was supported by NIH grant 1R01AI165570 and University of Illinois start-up funding to B.M.S, and the Lowell P. Hager Fellowship in Biochemistry to M.L.

## Author contributions

The study was conceived by B.M.S and M.L.; experiments were conducted by M.L. and A.G.M.; all authors contributed to data analysis and manuscript writing.

## Competing interests

BMS and ML are listed as inventors on a patent application that includes the design, production, and use of chimeric antibodies, some of which include teleost Ig heavy chain motifs. The remaining authors declare no competing interests.

## Additional information

Present address: Takeda Pharmaceuticals, Cambridge, MA 02139, USABeth M. Stadtmueller.

