## [Peer Review File · Nature Communications]

The structure of the teleost Immunoglobulin M core provides insights on polymeric antibody evolution, assembly, and functionEditorial Note: Parts of this Peer Review File have been redacted as indicated to remove third-party material where no permission to publish could be obtained.

REVIEWER COMMENTS

Reviewer #1 (Remarks to the Author):

This work is quite interesting, revealing structural patterns of multimeric immunoglobulins (Ig) throughout evolution. These data in the bony fish are certainly useful as a jumping off point for other studies of isotypes in jawed vertebrates. The data are convincing and interesting. There is a major omission in this work, however. The description of evolution of multimeric Igs on pp 3, 4, 12, 13, 19, 20, 22 are incomplete; this detracts from the work in a comparative sense, and readers will get the wrong impression of what is known in the field.

P3, line 42: J chain is also incorporated into IgM in amphibians (Hadji-Azimi, Immunology 1979, 30:187; note there are also EM pics showing hexamers—it is not known whether IgM is incorporated into pentamer and hexamer, but there are two high mw forms on non-reducing gels: Hsu, 1985, J. Immunol. 135:1998). It's been known for a long time that sharks incorporate J chain into multimeric IgM (Clem, Immunochemistry, 1976, 13:479 and Biochem. Biophys. Acta, 1976, 446:536, repeated by Steiner with a monoclonal antibody, J. Immunol., 2003, 170:6016, and two populations of plasma cells, one with J chain and the other without J chain, Flajnik, Eur. J. Immunol, 2013, 43:3061). J chain and Igs first appear in gnathostomes. The authors may have missed these pubs because they are rather old, but the work has been repeated in recent times.

P4, likewise, there are things known about multimeric Igs in ectotherms. Line 62 says that teleosts do not encode a J chain; this is true, but it should be mentioned that teleosts **have lost** the J chain (Venkatesh, Nature, 2014, 305:174), not that it had not yet emerged. Bony fish have many unusual characteristics (e.g., loss of MHC linkage associations, loss of J chain, emergence of bony fish-specific genes), some of which might be attributed to a teleost-specific genome-wide duplication (Venkatesh, Ann. Rev. Anim. Biosci. 2018, 6:47).

P4, Line 67, 'apparently' can be removed as it is clear there is no J chain associated with teleost IgM or IgT, as there is no J chain gene.

P12, IgM has been lost in the Coelocanth (Amemiya, Nature 2013, 496:311)

P13, it is said that the Tp is moderately conserved in vertebrates, but actually it is very well conserved (see below, Rumpf, J. Immunol. 2004, 173:1129), with the cysteine, size, and glycosylation site.

Canonical secretory tail

```
U51450 Gc IgW //KPNLVNVSLVLTESFNSCS/
1117935 Cp IgW //T-----K---/
X9 Re IgW //T-----D--K--V/
AY524298 Rp IgNAR //G--T-----DN-K--V/
Y13253 Ab IgM //G-QT----G-T-PDKA-F--N
U12456 Ac IgM //G--T---LN-KVPD//T-RG
M92851 Gc IgM TVDKSSG--SF--I--A-MDTI---Q/
70055 Hf IgM TVNKSSG--SF--I--A-LDTV---Q/
M29677 Re IgM TVNKSSG--SF-----MDTV---Q/
AAC12920 Hc IgM AVNKCSG--SF--I---QMDTV-A-Q/
U18701 Gc IgNAR //SS----V--SDTVK--T/
AF437724 PaeIgM //G--SS---TVIMSDTAGAY//
4467842 Hs IgM //G--T-Y-----MSDTAGT-Y/
AAB59662 Mm IgA //G--TN-S--VIMS-GDGI-Y/
546799 Hs IgA2 //G--TH----V-MA-VDGT-Y/
AF437727 PaeIgW //G--TT--L-VY---GKRE-HV
```

P19, mammalian IgM without the J chain is supposed to form hexamers, but in J chain KO mice, the IgM forms multimers of various sizes (Leanderson, *Eur. J. Immunol.* 1998, 28:2355), so still a lot to learn in vivo, which is probably the best place to look rather than in transfected cell lines.

P19, I'm not sure why the experiment with IgG tail was done—the IgG Tp is quite different from the IgM/IgA Tp—I might be missing something here. The other comparisons and chimeras are interesting.

P20, same as above on P3. J chain is incorporated into IgM of sharks and amphibians (no doubt). However, it does not seem to be incorporated into the multimeric amphibian IgX (Hsu, *Eur. J. Immunol.* 1996, 26:2833--also note that the IgX Tp lacks the Tp cys). Also, in sharks, monomeric IgM is a major component of serum and is induced in immunized animals—it is a product of plasma cells lacking the expression of J chain (Flajnik noted above). Why the shark IgM does not form multimers in the absence of J chain is not known, but this is true of shark IgNAR as well, i.e., IgNAR has the canonical Tp but does not form multimers. Still much to learn, but much **is known**.

P22, finally, there are teleost-specific features of IgM that should be considered. These features might help to explain some of the (beautiful) structural characteristics reported here. After immunization, there is a change in the nature of polymerization in IgM, apparently unique among vertebrates (Kaattari, *J. Immunol.* 2010, 184:844). The authors should have a look to see how their data might be reconciled with this functional work. Also unique among vertebrates, teleost cell surface IgM splices the CH3 domain to the transmembrane region instead of CH4 (found from sharks through mammals). Perhaps modification of teleost CH4 in the secretory IgM made it necessary to produce a novel transmembrane form (Warr, *Nucl. Acids. Res.*, 1990, 18:5227)?

Reviewer #2 (Remarks to the Author):

In this manuscript, the authors report the cryo-electron microscopy structure of the immunoglobulin M from a teleost species (tlgM). The structural characteristics of tlgM were analyzed in detail. The tlgM structure is terameric and has no JC domain, which is quite different from that of human IgM. Together with comparative and mutational analyses, the tlgM structure has provided the first glimpse of both a teleost plg and a plg lacking a JC, revealing distinct modes of assembly compared to mammalian plgs. Overall, this manuscript is well written, the experiments are well designed, and the quality of the data is high. The results would provide new insights into evolution and diversity of vertebrate plg structure and function. But this manuscript still has some issues that need to be revised as listed below.

1. Supplementary fig.1 panel b.

What does the lane L in Supplementary Figure 1 panel b mean? It should be the meaning of maker?

2. Line 107: Despite C2 biological symmetry..., what is the C2 mean? And “an axis running between between Fc μ 2...” this sentence repeats a word “between”.

3. Supplementary figures in the article are not cited in order. For example: Supplementary Figure 4 (line 227) should be cited first, then Supplementary Figure 5.

4. The residues Lys417 mentioned on line 149 is not shown in Figure 3-b or -c?

5. Lines 183-184: the cited figures are not only Supplementary Figure 5b, but also Supplementary Figure 5d and 5e; Line 190: Supplementary Figure 5c should be changed to 5f.

6. Figures should be cited in the order of Figures 5a, 5b, 5c, 5d (Line 217, 221, 225, and 232).

7. Line 260: Supplementary Figure 5c should be changed to 5f.

8. We know that the constant domain CH2 and Fabs of human IgM are flexible and poorly resolved in cryo-EM maps. During the construction of tlgM, whether do the authors try to construct the full-length tlgM (adding CH2 domain or Fabs)?

9. Line 510: it would be more appropriate to change the title to “Atomic model building, refinement, and validation”.

Reviewer #3 (Remarks to the Author):

Oligomerization of some immunoglobulins is thought to play a role in a wide range of functions. Despite their prevalence in vertebrates little is known about oligomerization processes, their evolution or the structure of those that lack a joining chain.

In this manuscript Lyu and co-workers present the structure of polymeric IgM from trout, which does not encode a joining chain. The structure shows that despite the fact that the eight heavy chains which form a tetramer of dimeric-IgM molecules have identical protein sequences, they adopt different folds. This results in a structure which is very distinct from the previously solved mammalian IgM oligomer structure. The authors then undertaken mutational analysis and analyze fusion proteins to provide data to support discussion on evolution and assembly.

The manuscript is generally of a very high level and contains exciting and interesting new data. The work supports the conclusions and the methodology is both sound and provide in sufficient detail.

However, there are some points which should be addressed. Specifically:

- 1) The authors refer to “electrostatics interactions” in several places e.g. lines 124, 149, when there are no charged residues involved.
- 2) It may be useful to have additional information e.g. RMSD as a supplementary table on the three distinct conformations of the six B-type domains. Linked to this in lines 217-219 it would be useful to know with which chains the RMSD was made and whether 1.27Å vs 1.18Å is really significantly different once the differences between the “three distinct conformations” of the B-type domains is considered?
- 3) Supplementary figure 1b provides supporting evidence that an inter-chain disulfide bond is formed, but this is not cited in the section relating to this possible disulfide. Adding a similar N/R SDS-PAGE gel for the C445S monomer would also support the arguments made.

4) The authors are in places a little loose with referring to their own figures. For example, supplementary figure 1 should be cited on line 92 not line 94; on lines 24-25 neither Fig 2a of supplementary figures 3a/3b show an interface “rich in electrostatic interactions”; 5d is an inappropriate citation on line 138; it should be 5h and not 5a on line 253 and 5f not 5c on line 260 – also 5c-d should possibly be cited around here in the text.

5) It may be useful for it to be reiterated to the reader around line 273 that the “domains differed, structurally but not in sequence, in the FG loop, G strand and pre-Tp loop regions”. Linked to this the section lines 282-313 could be potentially misleading to the casual reader as it has the implication that the sequences from tFcu A-type and B-type structures are different. I would suggest that the section (and the corresponding part of the discussion) be slightly reworded to make it clear this is not the case. This may also be supported by an alignment from say the start of strand F to the end of the protein for the wild-type and chimeric proteins used in Figure 6 (this could be a supplementary figure). This information is already included in Figure 5d, but the alignment is too small to read and requires the reader to work out where the boundaries are in the aligned sequences between the different chimeras.

Minor comments/suggestions:

Line 49, should it be “basolateral” rather than “basal lateral”?

line 66, insert “a” to make “which has a distinct”.

Line 179, insert “bonds” to make “potential disulfide bonds between”.

Line 193 opening sentence would benefit from a reference.

Lines 545-547, it may be useful to the reader to add common names of species as used in supplementary figure 4 to the names currently used in the text e.g. “*Oncorhynchus mykiss* (trout)”.

While readers could construct them themselves it may be useful to have all of the protein sequences in the supplementary material as well as an alignment of the native and chimeric proteins used in figure 6.

The font size of the proteins sequences in Figure 5d and supplementary figure 4 would greatly benefit from being a larger, readable size.

Supplementary figure 1, the legend talks about an N-terminal signal sequence, but this is

absent from the figure.

Supplementary figure 1, the legend talks about “incomplete reduction” for the reduced sample but it does not say anything about the high mw bands in the N lane nor that the gel strongly implies that the non-reduced SDS-treated sample is a disulfide linked dimer. I would suggest that the later get added and that either the extra bands in both lanes get mentioned or that the extra bands in both lanes do not get mentioned (not the current mix).

Responses to reviewer #1 comments:

Reviewer comment: *This work is quite interesting, revealing structural patterns of multimeric immunoglobulins (Ig) throughout evolution. These data in the bony fish are certainly useful as a jumping off point for other studies of isotypes in jawed vertebrates. The data are convincing and interesting. There is a major omission in this work, however. The description of evolution of multimeric Igs on pp 3, 4, 12, 13, 19, 20, 22 are incomplete; this detracts from the work in a comparative sense, and readers will get the wrong impression of what is known in the field.*

Response: We thank the reviewer for the positive feedback and construct comments, which we have incorporated into the revised manuscript. Specifically, we have endeavored to broaden discussion and reference to the evolution of multimeric Igs.

Reviewer comment: *“P3, line 42: J chain is also incorporated into IgM in amphibians (Hadji-Azimi, Immunology 1979, 30:187; note there are also EM pics showing hexamers—it is not known whether IgM is incorporated into pentamer and hexamer, but there are two high mw forms on non-reducing gels: Hsu, 1985, J. Immunol. 135:1998). It’s been known for a long time that sharks incorporate J chain into multimeric IgM (Clem, Immunochemistry, 1976, 13:479 and Biochem. Biophys. Acta, 1976, 446:536, repeated by Steiner with a monoclonal antibody, J. Immunol., 2003, 170:6016, and two populations of plasma cells, one with J chain and the other without J chain, Flajnik, Eur. J. Immunol, 2013, 43:3061). J chain and Igs first appear in gnathostomes. The authors may have missed these pubs because they are rather old, but the work has been repeated in recent times.”*

Response: Thank you for the comments. We limited and/or omitted discussion of amphibians and cartilaginous fish from the prior version owing to the manuscript being over length; however, we are happy the reviewer has emphasized the importance of including this information. Accordingly, we added several sentences to the introduction describing the heavy chain classes that can be assembled into pIgs in different species and also acknowledge the existence of JC in cartilaginous fish (Hohman, V. S. et al., J Immunol 170:6016, 2003) and amphibians (Hadji-Azimi, I. & Micheahamzhepour, M., Immunology 30:587, 1976; also see Castro, C. D. & Flajnik, M. F. The Journal of Immunology 193:3248, 2014). We have also modified the Introduction and Discussion sections to emphasize that JC is found in the genomes of all Gnathostomes except for teleosts.

Reviewer comment: *“P4, likewise, there are things known about multimeric Igs in ectotherms. Line 62 says that teleosts do not encode a J chain; this is true, but it should be mentioned that teleosts have lost the J chain (Venkatesh, Nature, 2014, 305:174), not that it had not yet emerged. Bony fish have many unusual characteristics (e.g., loss of MHC linkage associations, loss of J chain, emergence of bony fish-specific genes), some of which might be attributed to a teleost-specific genome-wide duplication (Venkatesh, Ann. Rev. Anim. Biosci. 2018, 6:47).”*

Response: As noted above, we have now stressed that teleosts have lost the JC gene in several parts of the manuscript.

Reviewer comment: *“P4, Line 67, ‘apparently’ can be removed as it is clear there is no J chain associated with teleost IgM or IgT, as there is no J chain gene.”*

Response: Thank you; we have implemented this change.

Reviewer comment: *“P12, IgM has been lost in the Coelocanth (Amemiya, Nature 2013, 496:311)”*

Response: Good point! This has been noted in the text and suggested reference has been included.

Reviewer comment: “P13, it is said that the *Tp* is moderately conserved in vertebrates, but actually it is very well conserved (see below, Rumpfelt, *J. Immunol.* 2004, 173:1129), with the cysteine, size, and glycosylation site.”

[Figure redacted]

Response: While we agree that there are conserved elements among *Tp* sequences, especially among residues that form the beta strand, there is variability among species including the number of C-terminal residues following the beta strand, their identity, and the position of the Cysteine residue relative to the C-terminus. We cannot rule out the possibility that these differences account for structural differences we observe between teleost and human IgM *Tps*. Given this, we have removed the term “*moderately conserved*” and replaced it with less subjective wording “... although residues near the C-terminus are variable, including the position of a conserved cysteine relative to the C-terminus (Supplementary Fig. 5).”

Reviewer comment: “P19, mammalian IgM without the *J* chain is supposed to form hexamers, but in *J* chain KO mice, the IgM forms multimers of various sizes (Leanderson, *Eur. J. Immunol.* 1998, 28:2355), so still a lot to learn *in vivo*, which is probably the best place to look rather than in transfected cell lines.”

Response: Indeed, multimers of multiple sizes have been reported in animals. Assuming this comment refers to lines 337-338 (now lines 344-347) which states “...*While this may reflect the higher degree of similarity shared between the B type fold and human IgM domains, the resulting plgs appeared to be tetramers rather hexamers, the predominant polymeric state associated with human IgM lacking the JC...*” The primary purpose of this sentence was to discuss data from our experiments, in which human IgM lacking the JC is predominantly hexameric as determined by SEC-MALS and our chimeric hFc μ -tFc μ is predominantly tetrameric as determined by SEC MALS (Fig. 6, Supplementary Fig. S6). This provides an “apples to apples” comparison reporting on how the sequence drives assembly in transfected cells; furthermore, while the JC KO mouse does express different sizes of polymers, the majority of literature on serum IgM lacking JC in humans suggests the predominant (but not only) form is hexameric. Accordingly, we have added a phrase at the end of the sentence, “..., *in our experiments (and also human serum)...*” (see line 347).

Reviewer comment: “P19, I’m not sure why the experiment with IgG tail was done—the IgG *Tp* is quite different from the IgM/IgA *Tp*—I might be missing something here. The other comparisons and chimeras are interesting.”

Response: In this experiment, chimeric Fcs utilized the human IgG Fc and teleost Tps. Because IgG is not normally assembled into plgs, we hypothesized that adding teleost Tps and observing the polymeric state could help shed light on how residues in the Tps, versus the residues in the Fc, contribute to the assembly mechanism.

Reviewer comment: “P20, same as above on P3. J chain is incorporated into IgM of sharks and amphibians (no doubt). However, it does not seem to be incorporated into the multimeric amphibian IgX (Hsu, Eur. J. Immunol. 1996, 26:2833--also note that the IgX Tp lacks the Tp cys). Also, in sharks, monomeric IgM is a major component of serum and is induced in immunized animals—it is a product of plasma cells lacking the expression of J chain (Flajnik noted above). Why the shark IgM does not form multimers in the absence of J chain is not known, but this is true of shark IgNAR as well, i.e., IgNAR has the canonical Tp but does not form multimers. Still much to learn, but much is known.”

Response: Thank you. We have added discussion stating that JC is incorporated into IgM in amphibians and sharks in our revised manuscript (see lines 332-333). We have also mentioned that IgX does not appear to incorporate JC (see line 358-360). We found it difficult to discuss IgX in detail or shark IgNAR, since they are different classes of antibodies and require considerable introduction (this is why there is relatively little discussion on teleost IgT and mammalian IgA in the original and revised version of the manuscript); furthermore comparable structural data are not available to use directly in analysis (as it is for human IgM). Finally, we found it difficult to add discussion on monomeric shark IgM in the absence of a published structure of any monomeric Fc with a Tp. This remains unknown for mammalian IgA as well, and our data on teleost IgM alone do not provide a clear explanation for how shark IgM (and human IgA) are able to form monomers in the absence of JC.

Reviewer comment: “P22, finally, there are teleost-specific features of IgM that should be considered. These features might help to explain some of the (beautiful) structural characteristics reported here. After immunization, there is a change in the nature of polymerization in IgM, apparently unique among vertebrates (Kaattari, J. Immunol. 2010, 184:844). The authors should have a look to see how their data might be reconciled with this functional work. Also unique among vertebrates, teleost cell surface IgM splices the CH3 domain to the transmembrane region instead of CH4 (found from sharks through mammals). Perhaps modification of teleost CH4 in the secretory IgM made it necessary to produce a novel transmembrane form (Warr, Nucl. Acids. Res., 1990, 18:5227)?”

Response: We thank the reviewer for this suggestion. Indeed, results presented in Kaattari, J. Immunol. 2010, 184:844 indicate that higher affinity tIgM antibodies have a higher degree of disulfide polymerization; this has implications for antibody assembly, stability and half-life. Kaattari et al suggest a possible model, in which the high-affinity antigen binding to BCRs is associated with a high degree of post-translational modification near the IgM C terminus, which promotes disulfide polymerization (and production of tIgM). This builds on earlier work described in Kaattari et al. Immunol. Rev. 1998, 166:133.

As far as we are aware, our manuscript is the first to conduct experiments specifically testing the role of tIgM Cys445. As we have shown in our manuscript, this cysteine residue found near the tIgM Tp C-terminus appears to be necessary for assembly (mutation results in a monomeric tFc_μ, see Supplementary Fig. 4g), and may covalently link copies of the heavy chain; however, density for the disulfide itself is poorly ordered in the structure (see Supplementary Fig. 4b-f). Taken together our data and Kaattari et al support the importance of C-terminal disulfide bonding at some stage(s) of tIgM assembly and/or function (and we have

now cited this paper in the revised discussion); however, interpretation beyond this is challenging without additional data.

Among these challenges (not discussed in the revised text) is the fact that disulfide formation and IgM tetramer formation has not been correlated in solution. For example, although Kaattari, *J. Immunol.* 2010, 184:844 show evidence for disulfide linked polymers using non-reducing composite acrylamide-agarose gels, it is not entirely clear if these represent equivalent polymeric species in solution (e.g., one disulfide stabilized dimer may non-covalently associate with another copy to form a tetramer that is stable in solution but dissociates on SDS-PAGE). Indeed, we observed a vast number of contacts (especially the interwoven Tps) that appear to have sufficient energy to stabilize a tetramer even in the absence of disulfide bonding. Furthermore, nonreducing SDS-PAGE (Supplementary Fig. 1) of our tFc μ tetramer reveals bands at the expected sizes of a monomeric and dimeric tFc μ , indicating that the tetramer is stable despite variability in disulfide bonding between heavy chains. Future experiments, beyond the scope of this work, may help better link Kaattari et al observations with assembly mechanisms and the structural data we present here. Furthermore, as we have noted in the text, the possibility that ER chaperones are involved in regulating the tIgM assembly process must also be explored (lines 385-386).

As noted by the reviewer, another curiosity of the two tCH μ 4 folding patterns is whether or not they would be compatible with the monomeric membrane-bound form of IgM (the B-cell receptor; IgM-BCR). Most jawed vertebrate species splice out the Tp to express the IgM-BCR; however, teleosts have lost a the splicing site and instead splice out all of IgM CH4 and link CH3 to the transmembrane domain (Ross DA et al., *Immunol Rev.* 1998; 166:143-151.). Thus, it appears that there is some advantage to removing CH4 from the membrane-bound form (presumably monomeric), but including it in secreted forms (tetrameric). As noted in our manuscript, the B-type tCH μ 4 fold is structurally more similar to mammalian IgM CH4 while the A-type fold results in different folding near the C-terminus. Speculatively, the A-type tCH μ 4 fold may not be compatible with direct linkage to the transmembrane domain and/or interactions with cellular membrane and/or interactions with signaling machinery. Another possibility is that if both the A-type and B-type CH4 domains are present during expression, the existence of two different folds may not be compatible with BCR assembly. Although we considered mentioning these possibilities in the revised text, ultimately, we omitted this topic because we felt it was quite speculative (and would require more discussion on BCR structure and function). Indeed, it seems difficult to fully consider these hypotheses without first understanding how/under which conditions it is that the fish folds CH4 into A-type and B-type domains. Additional experiments will be needed to address this curiosity!

Reviewer #2 (Remarks to the Author):

In this manuscript, the authors report the cryo-electron microscopy structure of the immunoglobulin M from a teleost species (tlgM). The structural characteristics of tlgM were analyzed in detail. The tlgM structure is tetrameric and has no JC domain, which is quite different from that of human IgM. Together with comparative and mutational analyses, the tlgM structure has provided the first glimpse of both a teleost plg and a plg lacking a JC, revealing distinct modes of assembly compared to mammalian plgs. Overall, this manuscript is well written, the experiments are well designed, and the quality of the data is high. The results would provide new insights into evolution and diversity of vertebrate plg structure and function. But this manuscript still has some issues that need to be revised as listed below.

Author Response: We thank the reviewer for the positive feedback and construct comments, which we have responded to below.

1. Supplementary fig.1 panel b.

What does the lane L in Supplementary Figure 1 panel b mean? It should be the meaning of maker?

Author Response: Thank you; indeed “L” designates the molecular weight standard and we have defined this in an updated figure legend.

2. Line 107: Despite C2 biological symmetry..., what is the C2 mean? And “an axis running between Fc μ 2...” this sentence repeats a word “between”.

Author Response: “C2” referred to two-fold rotational symmetry within the structure (that the tetramer is effectively a dimer with each half containing two Fcs); however, since this may be confusing to the audience, we have deleted the sentence and reworded the description.

3. Supplementary figures in the article are not cited in order. For example: Supplementary Figure 4 (line 227) should be cited first, then Supplementary Figure 5.

Author Response: Thank you for pointing this out, we have reviewed all the figures, the order in which they are cited, and corrected errors. Order of Supplementary Figs. 4 and 5 have been switched compared to the previous submission.

4. The residues Lys417 mentioned on line 149 is not shown in Figure 3-b or -c?

Author Response: Thank you; This was a typo and should have read Lys416. It has been corrected in the text (see line 156).

5. Lines 183-184: the cited figures are not only Supplementary Figure 5b, but also Supplementary Figure 5d and 5e; Line 190: Supplementary Figure 5c should be changed to 5f.

Author Response: Thank you for pointing this out, we have reviewed and corrected errors in the figures, the order in which they are cited, and the panels that are cited.

6. Figures should be cited in the order of Figures 5a, 5b, 5c, 5d (Line 217, 221, 225, and 232).

Author Response: Thank you for pointing this out; panels in Figure 5 are now cited in order.

7. Line 260: Supplementary Figure 5c should be changed to 5f.

Author Response: Thank you for pointing this out; Supplementary Fig 1b and Supplementary Fig 4g are cited.

8. We know that the constant domain CH2 and Fabs of human IgM are flexible and poorly resolved in cryo-EM maps. During the construction of tIgM, whether do the authors try to construct the full-length tIgM (adding CH2 domain or Fabs)?

Author Response: Thus far, our attempts to express full-length tIgM have been unsuccessful; however, we anticipate that alternative approaches may resolve this limitation in the future. In attempting expression of full-length tIgM, we obtained heavy chain and light chain nucleic acid sequence from two separate publications. The variable domains encoded by these two sequences presumably had different antigen specificity and may not have been able to form an intact Fab when co-transfected. Furthermore, we were unable to obtain antibodies recognizing teleost IgM and therefore our expression constructs contained N-terminal affinity tags to facilitate purification; the inclusion of these tags may have been unfavorable for the folding of the heavy and/or light chains. While we anticipate that full-length tIgM will be important for studying interactions with antigens and with Fc receptors, we do not anticipate that the inclusion additional domains will alter the structure of the Fc region (reported in this manuscript).

9. Line 510: it would be more appropriate to change the title to “Atomic model building, refinement, and validation”.

Author Response: We have changed this to “Atomic model building, refinement and validation.

Reviewer #3 (Remarks to the Author):

Oligomerization of some immunoglobulins is thought to play a role in a wide range of functions. Despite their prevalence in vertebrates little is known about oligomerization processes, their evolution or the structure of those that lack a joining chain.

In this manuscript Lyu and co-workers present the structure of polymeric IgM from trout, which does not encode a joining chain. The structure shows that despite the fact that the eight heavy chains which form a tetramer of dimeric-IgM molecules have identical protein sequences, they adopt different folds. This results in a structure which is very distinct from the previously solved mammalian IgM oligomer structure. The authors then undertaken mutational analysis and analyze fusion proteins to provide data to support discussion on evolution and assembly.

The manuscript is generally of a very high level and contains exciting and interesting new data. The work supports the conclusions and the methodology is both sound and provide in sufficient detail.

However, there are some points which should be addressed. Specifically:

Author Response: We thank the reviewer for the positive review of our work and have implemented the suggestions below into the revised manuscript.

1) The authors refer to “electrostatics interactions” in several places e.g. lines 124, 149, when there are no charged residues involved.

Author Response: Thank you; we have reviewed use of “electrostatic interactions” throughout the manuscript and reworded for clarity. In some cases, there are charged residues with potential to interact over longer distances (e.g. greater than 4Å) but other polar interactions at shorter distances and we have attempted to be more rigorous on the chemical descriptions of these interfaces (e.g. changed the wording to “polar interactions”).

2) It may be useful to have additional information e.g. RMSD as a supplementary table on the three distinct conformations of the six B-type domains.

Linked to this in lines 217-219 it would be useful to know with which chains the RMSD was made and whether 1.27Å vs 1.18Å is really significantly different once the differences between the “three distinct conformations” of the B-type domains is considered?

Author Response: Thank you for the comment; we have clarified how RMSDs were calculated in the text and methods and have added a related figure panel (revised Supplementary Fig. 4a) in the revised manuscript.

In the text we defined A-type vs. B-type folds based on structural differences in the CH4 domains. Excluding the Tps, the CH3 and CH4 domains in the six B-type chains adopt the same conformations. The exception to this is the difference between domains having an A strand and those having A and A' strands. As noted in the revised text, CH4 domains in chains B, C, F, G each have an A strand (Ig nomenclature) whereas those in chains A, D, E, and H have an A strand and an A' strand (where three residues following the A strand are able to H-bond with the adjacent B strand. Structurally, the associated difference in residue C-alpha position is

approximately 2-3 Å for residues 329-331 (in β strand A') and 4-6 Å for residues 331-335 (in the loop that precedes the B strand). Given this relatively small difference we did not note this in the previous version of the text; however, we have mentioned this difference in the revised manuscript. The 1.27Å vs 1.18Å RMSD values referred to above resulted from alignment of a human CH μ 4 (PDB ID 6KXS, chain A) domain to tFc μ CH4 domains from chains A and B, respectively (excluding the Tps residues). To determine if using other chains in the alignment markedly altered the RMSD values, we aligned human CH μ 4 to each CH4 domain in the refined tFc μ (see Table I below) and determined that human CH μ 4 is more similar to all six CH4 from B-type folds than to CH4 from A-type folds.

	human SIgM-Fc chain A (PDB ID: 6kxs)
chain A	1.386
chain B	1.047
chain C	1.048
chain D	1.075
chain E	1.048
chain F	1.052
chain G	1.025
chain H	1.218

Table 1: RMSD between indicated chains of tFc μ CH4 and hSIgM CH4 domains (chain A).

Accordingly, we have just reported the RMSDs for chains A and B (1.386 and 1.047, respectively), updated from the previous structural alignment that utilized an earlier build of the structure (see lines 225-228). We have not included the above table in the revised text since the overall result is unchanged from the original manuscript.

In the context of comparisons that include the tFc μ CH4 domains WITH Tps, symmetry-related chains (structurally equivalent chains from each half of the tetramer) adopt the same conformations, which we have now shown in revised Supplementary Fig. 4a. The RMSD calculations between symmetry-related chains we calculated using CH3-CH4-Tp (residues 225-445; also described in revised Methods). While RMSD is a useful means to quantify structural differences, in this case, viewing the alignments themselves and observing similarities and differences between A-type and B-type and human and teleost domains (Fig. 4c) and similarities between biological symmetry-related chains (Supplementary Fig. 4a) is likely the best way for the audience to visualize the comparisons.

3) Supplementary figure 1b provides supporting evidence that an inter-chain disulfide bond is formed, but this is not cited in the section relating to this possible disulfide. Adding a similar N/R SDS-PAGE gel for the C445S monomer would also support the arguments made.

Author Response: Thank you; we have reviewed the text and mentioned that our SDS-PAGE supports an inter-chain disulfide bonding. Additionally, we have modified Supplementary Fig. 4g to include a new SDS-PAGE gel, in which C445S tFc μ is run under both non-reducing and reducing conditions. This gel reveals no disulfide bonded chains. Two bands close to the same molecular weight are visible; given the proximity of Cys445 to the conserved N436 N-linked glycosylation site, we speculate that the mutation has promoted expression of two glycoforms.

As noted in the original manuscript, putative heterogeneous glycosylation is also observed on a subset of chimeric proteins.

4) The authors are in places a little loose with referring to their own figures. For example, supplementary figure 1 should be cited on line 92 not line 94; on lines 24-25 neither Fig 2a of supplementary figures 3a/3b show an interface “rich in electrostatic interactions”; 5d is an inappropriate citation on line 138; it should be 5h and not 5a on line 253 and 5f not 5c on line 260 – also 5c-d should possibly be cited around here in the text.

Author Response: Thank you for pointing this out, we have reviewed and corrected errors in the figures, the order in which they are cited, and the panels that are cited. As noted in our response to reviewer 2; we have also reviewed and, as needed, corrected the chemical description of interfaces shown in figures.

5) It may be useful for it to be reiterated to the reader around line 273 that the “domains differed, structurally but not in sequence, in the FG loop, G strand and pre-Tp loop regions”. Linked to this the section lines 282-313 could be potentially misleading to the casual reader as it has the implication that the sequences from tFc μ A-type and B-type structures are different. I would suggest that the section (and the corresponding part of the discussion) be slightly reworded to make it clear this is not the case.

Author Response: Thank you; this is a good point. We have added text (lines 279-281) reminding the audience that the sequences are the same.

Minor comments/suggestions: (**Author Responses in BOLD**)

Line 49, should it be “basolateral” rather than “basal lateral”? **Corrected.**

line 66, insert “a” to make “which has a distinct”. **Noted.**

Line 179, insert “bonds” to make “potential disulfide bonds between”. **Corrected.**

Line 193 opening sentence would benefit from a reference. **Corrected.**

Lines 545-547, it may be useful to the reader to add common names of species as used in supplementary figure 4 to the names currently used in the text e.g. “*Oncorhynchus mykiss* (trout)”. **Added.**

While readers could construct them themselves it may be useful to have all of the protein sequences in the supplementary material as well as an alignment of the native and chimeric proteins used in figure 6.

Thank you for these suggestions; we have added “Supplementary File 1” that contains a color-coded amino acid sequence of all 12 chimeras in this study.

The font size of the proteins sequences in Figure 5d and supplementary figure 4 would greatly benefit from being a larger, readable size. **Font size has been enlarged.**

Supplementary figure 1, the legend talks about an N-terminal signal sequence, but this is absent from the figure. **The caption of Supplementary figure 1a was changed to “schematic of the mature tFc μ domain organization” to be consistent with what is shown in the figure.**

Supplementary figure 1, the legend talks about “incomplete reduction” for the reduced sample but it does not say anything about the high mw bands in the N lane nor that the gel strongly implies that the non-reduced SDS-treated sample is a disulfide linked dimer. I would suggest that the later get added and that either the extra bands in both lanes get mentioned or that the extra bands in both lanes do not get mentioned (not the current mix). **Thank you; we have updated text regarding the disulfide-linked dimer as well as the Labels and descriptions in Fig. S1b, (and Fig. S4g).**

REVIEWERS' COMMENTS

Reviewer #1 (Remarks to the Author):

Thank you for making the suggested changes.

Reviewer #2 (Remarks to the Author):

The authors have addressed my concerns properly, and the absence of the joining chain in teleost IgM is interesting.

Reviewer #3 (Remarks to the Author):

This revised version of the manuscript addresses most of the concerns raised in the original submission.

I still have a concern over the use of the term "electrostatic interactions" in two places in the revised text. While all non-covalent interactions are ultimately electrostatic, the term is usually reserved for interactions between residues that have a fully charged group (rather than say an aromatic dipole). Line 131 refers to electrostatic interactions in Sup Fig 3b,c but no amino acids with charged side chains at physiological pH are shown. Similarly in figure 5 the green triangles which are said to indicate "hydrogen bonds and electrostatic interactions" do not label any amino acids that typically have charged side chains at physiological pH (except possibly one His). If the orientation of the aromatic side chains warrants it then "hydrogen bonds and dipole-dipole interactions" could be an alternative labeling.

Minor issues:

I could not find a reference in the text to supplementary file 1 or to supplementary figure 5.

Misplaced commas on lines 73 and 76

Line 102 refers to Supplementary figure 2, but the associated text does not appear to link to this figure

line 297 refers to Supplementary table 3, but the table does not appear to contain anything linked to the sentence it concludes.

PDB structures are cited on lines 466 and 526, but no reference is given

REVIEWERS' COMMENTS

Reviewer #1 (Remarks to the Author):

Thank you for making the suggested changes.

Reviewer #2 (Remarks to the Author):

The authors have addressed my concerns properly, and the absence of the joining chain in teleost IgM is interesting.

Reviewer #3 (Remarks to the Author):

This revised version of the manuscript addresses most of the concerns raised in the original submission.

I still have a concern over the use of the term "electrostatic interactions" in two places in the revised text. While all non-covalent interactions are ultimately electrostatic, the term is usually reserved for interactions between residues that have a fully charged group (rather than say an aromatic dipole). Line 131 refers to electrostatic interactions in Sup Fig 3b,c but no amino acids with charged side chains at physiological pH are shown. Similarly in figure 5 the green triangles which are said to indicate "hydrogen bonds and electrostatic interactions" do not label any amino acids that typically have charged side chains at physiological pH (except possibly one His). If the orientation of the aromatic side chains warrants it then "hydrogen bonds and dipole-dipole interactions" could be an alternative labeling.

Author Response: Thank you. We have revised text associated with the content of Supplementary Figure 3b, c. For Figure 5, we have changed the term "hydrogen bonds and electrostatic interactions" to "hydrogen bonds, electrostatic, or dipole-dipole interactions" to cover all possible interactions mediated by residues indicated by green or dark brown triangles. It was our intention that the green and brown triangles indicate the same possible interactions for intra-Fc and inter-Fc contacts, respectively, and there are more electrostatic interactions indicated by the brown triangles; this new wording should correctly apply to both.

Minor issues:

I could not find a reference in the text to supplementary file 1 or to supplementary figure 5.

Author Response: Thank you. Supplementary file 1 has now been cited in the Figure 6 legend (but has been transformed into supplementary table 5. Supplementary figure 5 was cited in line 237 and is still cited in the revised text.

Misplaced commas on lines 73 and 76.

Author Response. Thank you. We are unsure about the use of commas in this sentence and have noted the location in the revised text (tracked changes) and suggest copy editors check the grammar.

Line 102 refers to Supplementary figure 2, but the associated text does not appear to link to this figure

Author Response: Supplementary Figure 2 was cited in line 104 because this sentence described overall quality of cryo-EM density map and Supplementary Figure 2e showed the overall and local resolution of cryo-EM map. In the most updated version, we cite only panel e.

line 297 refers to Supplementary table 3, but the table does not appear to contain anything linked to the sentence it concludes.

Author Response: Thank you for catching this. Supplementary Table 4 is the correct table to cite here and we have fixed this in the most updated version.

PDB structures are cited on lines 466 and 526, but no reference is given

Author Response: References are now added.